# Soluble pathogenic tau enters brain vascular endothelial cells and drives cellular senescence and brain microvascular dysfunction in a mouse model of tauopathy

Stacy A. Hussong[1,2,3,14], Andy Q. Banh[4,5,14], Candice E. Van Skike[5], Angela O. Dorigatti[5], Stephen F. Hernandez [5], Matthew J. Hart[1,2,6], Beatriz Ferran[1,2], Haneen Makhlouf[1,2], Maria Gaczynska [5,7], Pawel A. Osmulski [5,7], Salome A. McAllen [8,9,10], Kelly T. Dineley[8,9,10], Zoltan Ungvari[2,11,12], Viviana I. Perez[13], Rakez Kayed [8,9,10] & Veronica Galvan [1,2,3] ✉

Vascular mechanisms of Alzheimer's disease (AD) may constitute a therapeutically addressable biological pathway underlying dementia. We previously demonstrated that soluble pathogenic forms of tau (tau oligomers) accumulate in brain microvasculature of AD and other tauopathies, including prominently in microvascular endothelial cells. Here we show that soluble pathogenic tau accumulates in brain microvascular endothelial cells of P301S(PS19) mice modeling tauopathy and drives AD-like brain microvascular deficits. Microvascular impairments in P301S(PS19) mice were partially negated by selective removal of pathogenic soluble tau aggregates from brain. We found that similar to trans-neuronal transmission of pathogenic forms of tau, soluble tau aggregates are internalized by brain microvascular endothelial cells in a heparin-sensitive manner and induce microtubule destabilization, block endothelial nitric oxide synthase (eNOS) activation, and potently induce endothelial cell senescence that was recapitulated in vivo in microvasculature of P301S(PS19) mice. Our studies suggest that soluble pathogenic tau aggregates mediate AD-like brain microvascular deficits in a mouse model of tauopathy, which may arise from endothelial cell senescence and eNOS dysfunction triggered by internalization of soluble tau aggregates.

Amyloid-β (Aβ) and pathogenic misfolded forms of tau protein have been causally implicated in the pathogenesis of Alzheimer's disease, the leading cause of dementia in the elderly. In its native form, tau is involved in microtubule stabilization[1]. Under pathological conditions, such as in 'pure' tauopathies or AD, hyperphosphorylated tau detaches from microtubules, destabilizing the microtubule cytoskeleton. The phosphorylation of tau frequently precedes tau multimerization, including oligomerization[2–4]. Tau oligomers (soluble prefibrillar tau aggregates) are a distinct pathogenic aspect of AD and other dementias[5–7] that trigger significant synaptic and cognitive deficits[8]. Consistent with recent studies showing synaptic dysfunction and cell death in tauopathies prior to neurofibrillary tangle formation[9], accumulation of tau oligomers correlates with the onset of clinical symptoms in AD[7].

Tau, including but not limited to its oligomeric form, is released extracellularly[10,11], and transfers trans-neuronally, including trans-synaptically, spreading in patterns that follow synaptic transmission throughout brain circuits as tauopathies progress[12,13]. This pattern of propagation along with robust in vitro evidence[14,15] have been posited as evidence for prion-like properties of misfolded tau, giving rise to a model in which tau oligomers released into the extracellular space are transmitted to neighboring neurons where upon uptake, they promote native tau phosphorylation, misfolding, and aggregation[12,14,16] with consequent destabilization of the microtubule cytoskeleton[1,3]. This sequence of events triggers a vicious cycle of formation of misfolded tau that is prone to aggregation[13]. Supporting the validity of this model, levels of extracellular tau in AD brain interstitial fluid are comparable or even higher than those of Aβ oligomers[11,17]. Thus, if the prion-like tau propagation model is correct, extracellular tau plays a key role in mediating pathogenesis of tauopathy and therefore understanding its impact in the central nervous system is critical.

While the original model for transcellular propagation of soluble tau aggregates is centered on neurons, tau oligomers also accumulate in the brain microvasculature, both in neurodegenerative diseases such as AD and progressive supranuclear palsy (PSP), as wells as in mouse models of AD[18]. This evidence suggests that oligomeric tau (heretofore also soluble tau aggregates) may contribute to cerebrovascular dysfunction, an early event in the etiology of AD[19,20] and other tauopathies[21]. Soluble tau aggregates co-localize with markers of brain microvascular endothelial cells[18], suggesting that tau released from neurons[10] may reach neighboring microvessels[18,22]. This is consistent with recent reports suggesting clearance of tau through the blood-brain barrier (BBB)[23,24] that is formed in part by cerebrovascular endothelial cells. Like neurons, endothelial cells also express tau[25,26] and while the trans-neuronal transmission of misfolded forms of tau has been well established, the potential impact of pathogenic tau on cerebrovascular endothelial cells has not yet been explored.

With age, many somatic cells accumulate damage that leads to cellular senescence, characterized by cell cycle arrest and the acquisition of a senescence-associated secretory phenotype (SASP)[27]. Accumulation of senescent cells thus creates a proinflammatory environment that compromises tissue function[28]. Endothelial cell senescence critically contributes to age-associated vascular dysfunction and pathology[29] by reducing nitric oxide (NO) bioavailability[30]. Brain vascular dysfunction is a critical mediator of AD pathogenesis[31,32] and a recent report suggested that cerebrovascular dysfunction may be the earliest and most abnormal biomarker in AD progression[20].

The present studies sought to define the impact of tau aggregate accumulation in brain microvascular endothelial cells on vascular function and were designed to test the hypothesis that soluble tau aggregates enter brain microvascular endothelial cells and trigger pathogenic processes leading to microvascular deficits of AD. To this aim we measured entry of pathogenic oligomeric tau in primary human brain endothelial cells (HBEC) and defined the impact of oligomeric tau entry on endogenous native endothelial cell tau phosphorylation, microtubule stability, and critical aspects of brain vascular endothelial cell function, both in vitro and in vivo in mutant human tau transgenic mice modeling tauopathy. The causal involvement of pathogenic soluble tau aggregates in AD-like microvascular deficits was defined using immunotherapy that specifically removed pathogenic tau aggregates from brain.

## Results

### Heparin-sensitive pathogenic soluble tau aggregate entry into primary human brain vascular endothelial cells induces endogenous endothelial cell tau phosphorylation

Pathogenic soluble tau aggregates (tau oligomers) accumulate in cortical capillaries and arterioles of AD and PSP brain[18] (secondary and primary tauopathies respectively) in association with brain microvascular endothelial cells[18]. To define whether accumulation of pathogenic aggregated tau in brain microvascular endothelial cells is recapitulated in a model of tauopathy we immunostained brain sections from mice modeling tauopathy that express aggregation-prone P301S-mutant human tau [P301S(PS19) mice]. Similar to our observations in AD and PSP brains, we found abundant soluble aggregates in association with cortical capillaries of P301S(PS19) mice (Fig. 1A) at 8 months of age. Tau aggregates, including oligomers[10], are released extracellularly and promote native tau phosphorylation and aggregation in neurons[12,16]. Previous reports have shown that pathogenic tau is internalized by neurons through micropinocytosis via tau binding to cell surface heparan sulfate proteoglycans (HSPG)[33,34]. To define whether accumulation of pathogenic aggregated tau in microvascular endothelial cells of AD and other tauopathies[18] and in models of tauopathy (Fig. 1A) may be a consequence of HSPG-mediated entry of pathogenic soluble aggregated tau into endothelial cells, we incubated human brain microvascular endothelial cells (HBEC) with epitope-tagged recombinant human tau soluble aggregates (Tau-V5) and competitively inhibited tau HSPG binding sites with heparin or treated cells with vehicle[33,34] (Fig. 1B). Similar to pathogenic tau entry into neurons[10,12,14,35], we found that tau entered primary HBEC, where it co-localized with microtubules (Fig. 1B). In agreement with studies of pathogenic tau uptake by neurons, we found that co-treatment with heparin almost completely blocked tau-V5 internalization in HBEC, indicating that tau entry into HBEC is HSPG-dependent. Soluble tau oligomers, but not tau monomers or human cytokeratin-8 (KRT8), an unrelated exogenous protein of similar size and charge as tau, promoted phosphorylation of endogenous HBEC tau ($p < 0.05$, Fig. 1C, D)[36]. Consistent with these observations and prior reports[26,37,38], we found that expression of tau in HBEC was comparable to that of human cells of neuronal origin (SK-N-SH neuroblastoma-derived cells $p = 0.14$, Supplementary Fig. 1A). Other human primary endothelial cells (human umbilical vein endothelial cells, HUVEC) also express tau (Supplementary Fig. 1A) and endogenous tau protein is readily detectable in HBEC by immunoblot (Supplementary Fig. 1B) and by immunocytochemistry, the latter showing endogenous tau in proximity to microtubules (Supplementary Fig. 1C). While both V5-tagged tau (Tau-V5) and untagged, purified soluble aggregated tau contained both monomers and low-order tau oligomers (Fig. 1E, F), monomeric forms of tau were much less abundant in purified soluble aggregated tau oligomer preparations (Fig. 1E, F), that also more closely resembled the diversity in size distribution of tau oligomers in brain[24]. Taken together, these data suggest that, similar to entry of pathogenic tau into neurons, internalization of soluble tau aggregates by human brain vascular endothelial cells leads to phosphorylation of endogenous tau.

### Pathogenic soluble tau aggregates trigger microtubule destabilization and block eNOS activation in primary human brain microvascular endothelial cells

Hyperphosphorylation of tau leads to its detachment from microtubules and destabilization of the microtubule cytoskeleton in neurons[1]. Consistent with these observations, we found that internalization of soluble tau aggregates reduced levels of acetyl-α-tubulin, a marker of microtubule stability[39] ($p < 0.001$, Fig. 2A, B), and decreased overall microtubule density ($p < 0.0001$, Fig. 2C, D) in HBEC. The impact on microtubule stability was specific to oligomeric tau, since levels of acetyl-α-tubulin did not change as a result of KRT8 nor monomeric tau exposure (Fig. 2A, B). Exposure to tau monomers, however, caused a significant decrease in microtubule density ($p < 0.0001$, Fig. 2C, D). This may be a consequence of tau oligomerization during incubation since we found that monomeric tau spontaneously formed oligomers in media collected from tau monomer-treated HBEC (Supplementary Fig. 2A, B). Indeed, after incubation with tau monomers, HBEC media contained tau aggregates that were similar in size distribution to tau oligomer preparations (Supplementary

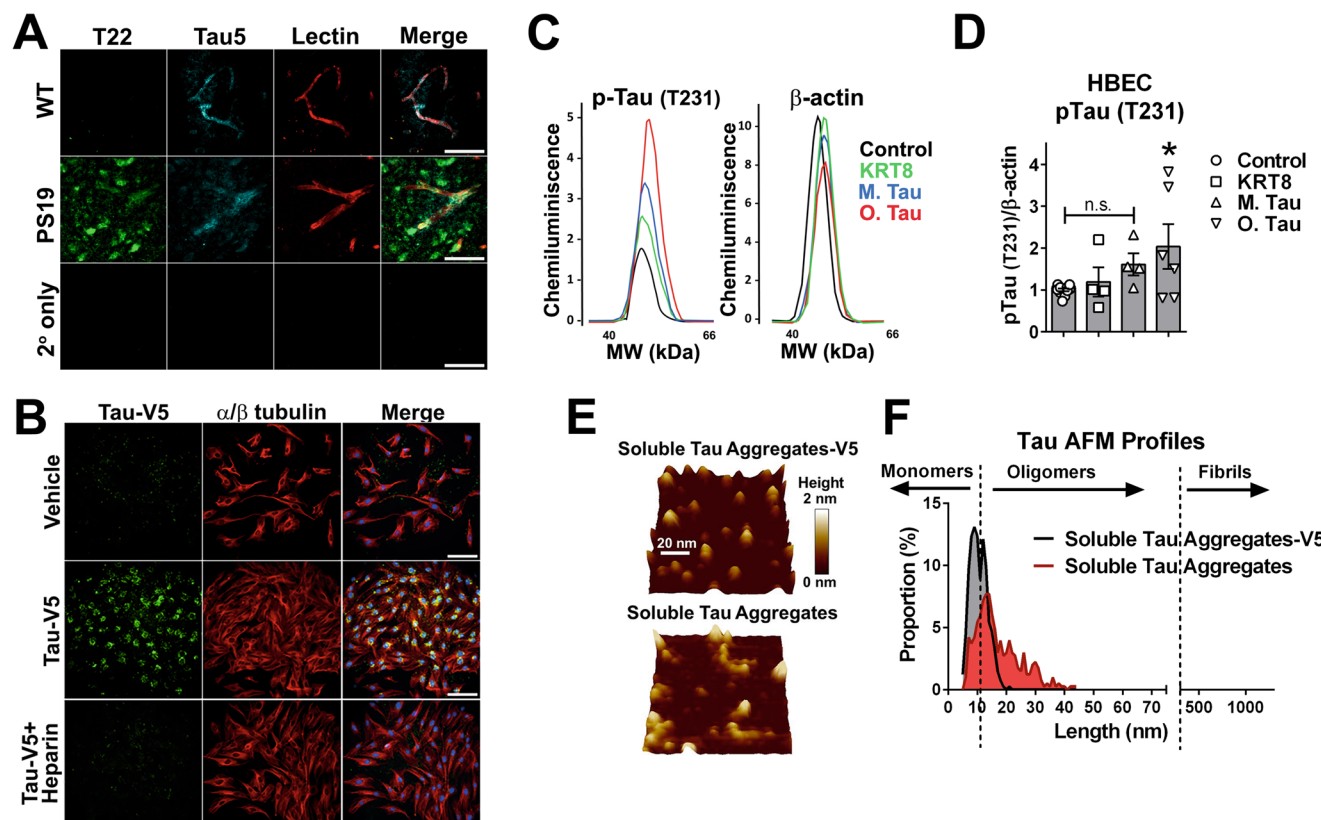

**Fig. 1 | Heparan sulfate proteoglycan (HSPG)-dependent internalization of soluble extracellular tau aggregates by brain microvascular endothelial cells triggers endogenous tau phosphorylation (Thr231). A–C** Soluble aggregated tau internalization triggers endogenous tau phosphorylation in primary human brain microvascular endothelial cells (HBEC). **A** Soluble tau aggregates accumulate in brain microvasculature of P301S(PS19) mice. Representative images of cortical brain sections from 8-month-old male and female P301S(PS19) and WT controls showing soluble tau aggregates (T22, green) and total tau (Tau5, cyan) immunoreactivity in lectin-stained microvasculature (red). Nine independent samples (three brain sections from each of three mice) were examined over three independent experiments. **B** Representative images of primary HBEC exposed to soluble aggregates of V5-tagged recombinant human tau-441 or vehicle (control) in the presence or absence of heparin, immunostained with antibodies for V5 (green) and α/β-tubulin (red), and counterstained with DAPI (blue). Scale bar is 200 μm. Nine independent samples were examined over 3 independent experiments. **C** Representative electropherograms

from capillary electrophoresis immunoassays for phosphorylated tau (T231) and β-actin in lysates from HBEC treated with recombinant human cytokeratin-8 (KRT8), monomeric tau protein (M. Tau), unlabeled soluble tau aggregates (O.Tau), or vehicle (control). **D** Quantitative analyses of data in **C** ($F_{(3,22)} = 3.115$, ANOVA, $p = 0.047$). Control vs. O. Tau, *, $P < 0.05$. Control, $n = 12$; KRT8, $n = 4$; M. Tau, $n = 4$, O. Tau, $n = 6$ biologically independent samples examined over 6 independent experiments. Data are means ± SEM. **E, F** Atomic force microscopy profiling of tau. **E** Representative AFM images of field fragments with soluble tau aggregates, tagged (V5) and untagged. **F** Size distribution of tau particles as measured by AFM show overlapping distributions of soluble tau aggregates (>12 nm in length) in non-tagged and V5-tagged soluble tau aggregate preparations (Soluble tau aggregates V5, $n = 405$ particles; soluble tau aggregates, $n = 403$ particles [i.e., biologically independent samples (particles measured) examined over 1 independent experiment]. Data are frequency profiles. For post-hoc analyses, lack of a specific $P$ value in the legend reflects the information reported by GraphPad Prism Version 9.4.0.

Fig. 2A, B). KRT8, although similar in size and charge to tau, did not oligomerize during incubation with HBEC since size distributions of KRT8 after incubation in media were similar to those of tau monomers prior to incubation (Supplementary Fig. 2A, C). These data suggest that decreases in microtubule density observed after exposure of HBEC to tau monomers may be a consequence of spontaneous oligomerization of tau monomers during incubation. Taken together, our studies indicate that soluble oligomeric tau aggregates trigger microtubule destabilization and decrease microtubule density in HBEC.

Endothelial nitric oxide synthase (eNOS) activation requires microtubule-dependent transport of the enzyme to membrane compartments[40]. To determine if microtubule destabilization following soluble pathogenic tau uptake by HBEC inhibits eNOS translocation to the cell surface we measured eNOS localization in HBEC stimulated with acetylcholine (ACh) after exposure to soluble tau oligomers, tau monomers, or vehicle control. Consistent with the requirement for the microtubule cytoskeleton in the transport of eNOS[39,41], we found that eNOS abundance at the cell periphery was significantly decreased in HBEC treated with oligomeric tau, where the

microtubule cytoskeleton is destabilized, as compared to controls ($p = 0.002$, Fig. 2E, F). eNOS localization to the perinuclear region was also reduced when HBEC were treated with tau oligomers or monomers, but this apparent difference was not significant ($p = 0.45$, Fig. 2E–G). The majority of active eNOS is located at the cell membrane (edge-associated signal, Fig. 2E, F)[42–45] and the endoplasmic reticulum (perinuclear region signal, Fig. 2E–G)[42]. Consistent with the decrease in eNOS at the cell periphery, we found that basal NO production was significantly decreased in HBEC exposed to either monomeric or oligomeric tau ($p = 0.0048$ or $p < 0.0001$, respectively, Fig. 2H). Furthermore, soluble tau aggregates significantly decreased NO production in HBEC as compared to tau monomers ($p = 0.0473$, Fig. 2H). eNOS activity is regulated through increased phosphorylation of the enzyme at Ser1177 (activating) coupled with a decrease in Thr495 (inhibitory) phosphorylation. Impaired eNOS-dependent NO release is central to endothelial dysfunction and can be elicited by alterations of the microtubule system[39,40,46]. To define whether microtubule destabilization following tau internalization by HBEC is associated with changes in eNOS activation, we quantitated

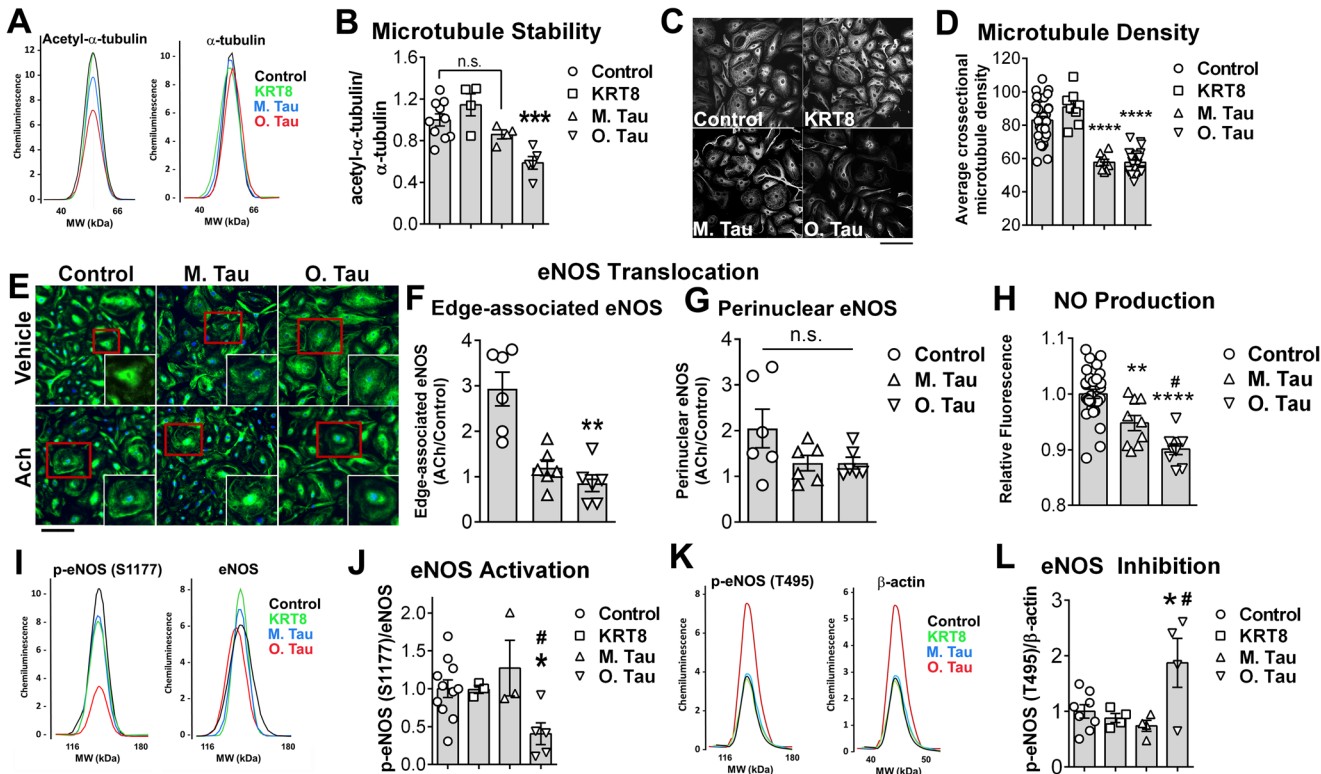

**Fig. 2 | Soluble tau aggregates destabilize microtubules, reduce microtubule density, and block eNOS activation in human brain microvascular endothelial cells. A, B** Soluble tau aggregates destabilize microtubules in primary human brain microvascular endothelial cells (HBEC). **A** Representative electropherograms from capillary electrophoresis immunoassays of acetyl-α-tubulin and α-tubulin in lysates of cells treated with recombinant human cytokeratin-8 (KRT8), monomeric tau protein (M. Tau), soluble tau aggregates (O.Tau), or vehicle (control). **B** Quantitative analyses of data represented in (**A**). (F(3,19) = 9.871, ANOVA, $P = 0.0004$, O. Tau different from Control by Tukey's test, ***$p < 0.001$. Control, $n = 11$; KRT8, $n = 4$; M. Tau, $n = 4$, O.Tau, $n = 5$ biologically independent samples examined over 6 independent experiments. Data are means ± SEM. **C, D** Soluble tau aggregates reduce microtubule density in HBEC. **C** Representative images of β-tubulin immunoreactivity in HBEC exposed to KRT8, M. Tau, O.Tau, or vehicle (control). Scale bar is 200 μm. **D** Quantitative analyses of microtubule density distributions from data in **C** ($H = 49.37$, $P < 0.0001$, Kruskal–Wallis. M. Tau and O. Tau different from Control by Dunn's ****$P < 0.0001$. Control, $n = 31$; KRT8, $n = 8$; M. Tau, $n = 8$, O. Tau, $n = 27$ biologically independent samples examined over three independent experiments. Data are representative images and means ± SEM. **E, F** Soluble tau aggregates inhibit eNOS translocation to the cell membrane. **E** Representative images of eNOS and β-tubulin immunoreactivity in HBEC exposed to KRT8, M. Tau, O.Tau and induced with acetylcholine (ACh) or vehicle (control). The scale bar is 200 μm. **F** Quantitation of edge-associated eNOS (H = 12.04, $p = 0.0002$, Kruskal-Wallis, O.Tau different from Control by Dunn's, **$p = 0.002$), $n = 6$ (all experimental groups) biologically independent samples examined over 2 independent experiments and **G** perinuclear-localized eNOS from images in E in ACh-induced cells normalized to the vehicle for each treatment group (H = 1.719,

$P = 0.4527$, Kruskal–Wallis, n.s., $n = 6$ (all experimental groups) biologically independent samples examined over 2 independent experiments. Soluble tau aggregates decrease nitric oxide (NO) production in HBEC. **H** Quantification of the relative fluorescence of DAF-FM, a nitric oxide indicator, in HBEC treated with M. Tau, O. Tau, or vehicle (Control) (F(2,45)=22.15, ANOVA, $p < 0.0001$; Control vs M. Tau, ** $p = 0.0048$; Control vs O.Tau, ****$p < 0.0001$, M. Tau vs O.Tau, #$p = 0.0473$ by Tukey's. Control, $n = 30$; M. Tau, $n = 9$, O. Tau, $n = 9$ biologically independent samples examined over three independent experiments. Data are means ± SEM. **I, J** Soluble tau aggregates decrease eNOS activation by phosphorylation at S1177. **I** Representative electropherograms from capillary electrophoresis immunoassays of phosphorylated eNOS (S1177) in HBEC treated with KRT8, M. Tau, O.Tau, or vehicle (control). **J** Quantitative analyses of data in G (F(3,18) = 3.903, ANOVA, $p = 0.027$, Control vs. O. Tau, *$p < 0.05$; M. Tau vs. O. Tau, #, $p < 0.05$; Control vs. M. Tau, n.s., from Tukey's. Control, $n = 11$; KRT8, $n = 3$; M. Tau, $n = 3$, O.Tau, $n = 5$ biologically independent samples examined over 6 independent experiments. Data are means ± SEM. **K, L** Soluble tau aggregates promote eNOS inhibition by phosphorylation at Thr495. **K** Representative electropherograms from capillary electrophoresis immunoassays of phosphorylated eNOS (Thr495) in HBEC treated with KRT8, M. Tau, O.Tau, or vehicle (Control). **L** Quantitative analyses of data in I (F(3,16)=5.079, ANOVA, $p = 0.012$, Control vs O.Tau, *$p = 0.03$; M. Tau vs O.Tau, #$p = 0.014$, Control vs. M. Tau, n.s., $p = 0.79$. Control, $n = 8$; KRT8, $n = 4$; M. Tau, $n = 4$, O.Tau, $n = 4$ biologically independent samples examined over four independent experiments. Data are representative electropherograms, and means ± SEM. For post-hoc analyses, lack of a specific $P$ value in the legend reflects the information reported by GraphPad Prism Version 9.4.0.

phosphorylation of eNOS at Ser1177 and Thr495. Phosphorylation of eNOS at Ser1177 was significantly reduced ($p < 0.05$, Fig. 2I, J) and phosphorylation at Thr495 was significantly increased ($p = 0.03$, Fig. 2K, L) in HBEC exposed to tau oligomers but not in HBEC exposed to KRT8 or monomeric tau as compared to controls. Together, these data indicate that soluble tau aggregate entry decreases eNOS activation and NO release in HBEC. Our data suggest that destabilization of the microtubule cytoskeleton precedes changes in eNOS activity, as the spontaneous accumulation of tau oligomers during incubation is sufficient to trigger changes in microtubule density (Fig. 2C, D), but not in eNOS activation state in tau-treated HBEC (Fig. 2I, J). Furthermore,

we found that the magnitude of the decrease in NO production arising from tau treatment is significantly larger in HBEC treated with tau oligomers as compared to tau monomers (Fig. 2H). Taken together, these data suggest that tau internalization blocks eNOS activation in HBEC by inhibiting eNOS transport to cellular membranes and activation through destabilization of microtubules.

**Pathogenic soluble tau aggregates induce senescence in primary human brain microvascular endothelial cells**

Disruption of the microtubule network has been associated with the activation of cellular senescence[47]. Consistent with this knowledge, we

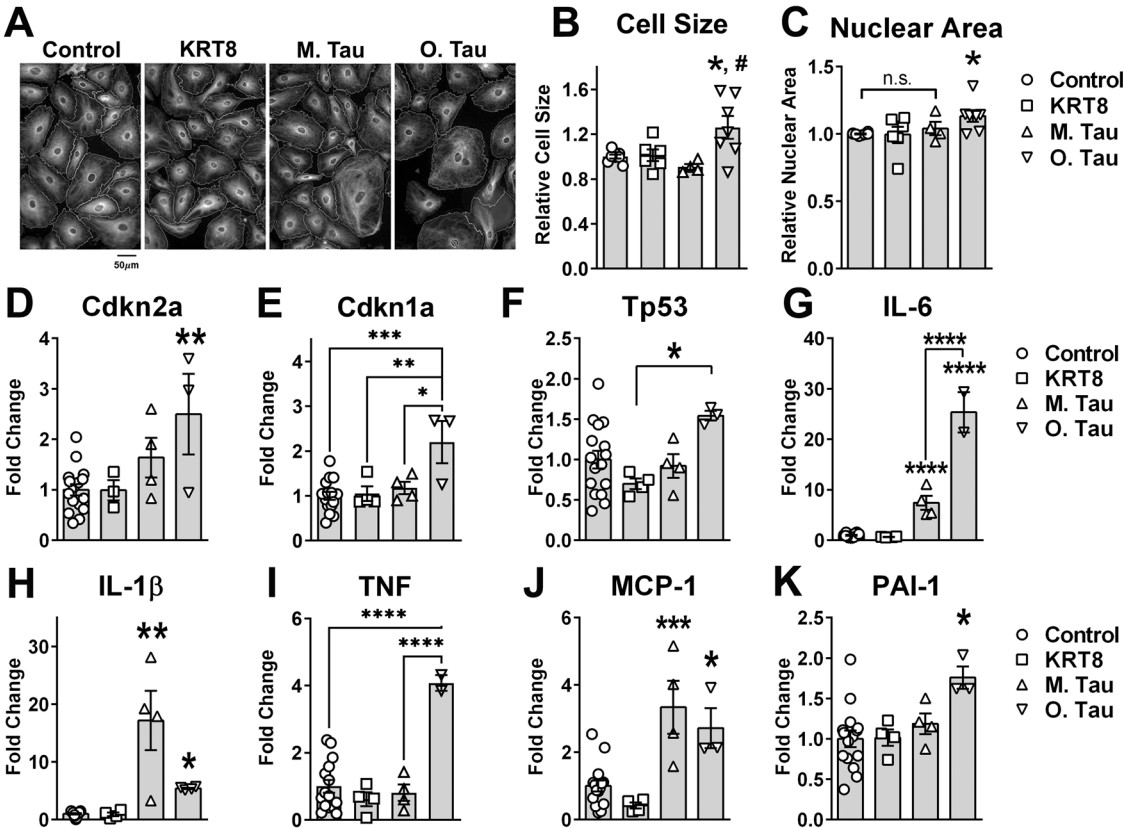

**Fig. 3 | Soluble tau aggregates induce senescence/SASP in primary human brain microvascular endothelial cells.** **A**–**C** Soluble tau aggregates increase cell and nuclear size in primary human brain microvascular endothelial cells (HBEC).
**A** Representative images of HBEC treated with recombinant human cytokeratin-8 (KRT8), monomeric tau protein (M. Tau), soluble tau aggregates (O.Tau), or vehicle (Control). **B** Quantitative analyses of cell size from data shown in A. (F(3,21)=5.391, ANOVA, $p = 0.0065$; Control vs. O. Tau, *, $p < 0.05$, M.Tau vs. O. Tau #. $p < 0.05$ by Tukey's. Control, $n = 8$; KRT8, $n = 6$; M. Tau, $n = 4$, O.Tau, $n = 7$ biologically independent samples examined over 2 independent experiments. Data are means ± SEM. **C** Nuclear area (H = 7.989, $p = 0.046$; Control vs. O. Tau, *$p = 0.025$, Control vs. M.Tau, n.s., $p > 0.999$ by Dunn's. Control, $n = 8$; KRT8, $n = 6$; M. Tau, $n = 4$, O.Tau, $n = 7$ biologically independent samples examined over 2 independent experiments. Data are representative images and means ± SEM. **D**–**F** Soluble tau aggregates trigger cell cycle arrest in HBEC. **D** Quantitative real-time PCR (qRT-PCR) measurements of mRNA abundance for cyclin dependent kinase inhibitor 2 A (Cdkn2a, (F(3,22)=5.264, ANOVA, $p = 0.007$, Control vs. O. Tau, **, $p < 0.01$, from Tukey's. Control, $n = 16$; KRT8, $n = 3$; M. Tau, $n = 4$, O.Tau, $n = 3$ biologically independent samples examined over 7 independent experiments. **E** Cyclin dependent kinase inhibitor 1 A (CDKN1A, F(3,23)=7.545, ANOVA, $p = 0.0011$, Control vs. O. Tau, ***$p < 0.001$; M. Tau vs. O.Tau, *$p < 0.05$; KRT8 vs. O. Tau, **$p < 0.01$ from Tukey's. Control, $n = 16$; KRT8, $n = 4$; M. Tau, $n = 4$, O.Tau, $n = 3$ biologically independent samples examined over 7 independent experiments). **F** Tumor protein 53 (Tp53, (F(3,23) = 3.050, ANOVA, $p = 0.049$, KRT8 vs. O. Tau, *$p < 0.05$ by Tukey's. Control, $n = 16$; KRT8, $n = 4$; M. Tau, $n = 4$, O.Tau, $n = 3$ biologically independent samples examined over 7 independent experiments). **G**–**K** Soluble tau aggregates promote expression of the senescence-associated secretory phenotype in HBEC.
**G** Interleukin 6 (IL-6, F(3,22) = 147.7, ANOVA, $p < 0.0001$. Control vs. O. Tau, ****$p < 0.0001$; Control vs. M. Tau, ****$p < 0.0001$; M. Tau vs. O. Tau, ****$p < 0.0001$ by Holm–Sidak's. Control, $n = 16$; KRT8, $n = 4$; M. Tau, $n = 4$, O.Tau, $n = 2$ biologically independent samples examined over 7 independent experiments). **H** Interleukin 1β (IL-1β, H = 16.49, Kruskal–Wallis, $p = 0.0009$. Control vs. O. Tau, *$p < 0.05$; Control vs. M. Tau, **$p < 0.01$. Control, $n = 15$; KRT8, $n = 4$; M. Tau, $n = 4$, O.Tau, $n = 4$ biologically independent samples examined over 7 independent experiments). **I** Tumor necrosis factor (TNF, F(3,21) = 14.85, ANOVA, $p < 0.0001$. Control vs. O. Tau, ****$p < 0.0001$; M. Tau vs. O. Tau, ****$p < 0.0001$ by Dunn's. Control, $n = 15$; KRT8, $n = 4$; M. Tau, $n = 4$, O.Tau, $n = 2$ biologically independent samples examined over seven independent experiments). **J** Monocyte chemoattractant protein-1(MCP-1, F(3,22) = 12.44, ANOVA, $p < 0.0001$. Control vs. O. Tau, *, $p < 0.05$; Control vs. M. Tau, ***$p < 0.01$ by Holm–Sidak's. Control, $n = 15$; KRT8, $n = 4$; M. Tau, $n = 4$, O.Tau, $n = 3$ biologically independent samples examined over seven independent experiments). **K** Plasminogen-activator inhibitor-1 (PAI-1, F(3,22) = 4.173, ANOVA, $p = 0.018$. Control vs. O. Tau, *$p < 0.05$ by Holm–Sidak's. Control, $n = 15$; KRT8, $n = 4$; M. Tau, $n = 4$, O.Tau, $n = 3$ biologically independent samples examined over seven independent experiments). Data are representative images and means ± SEM. For post-hoc analyses, lack of a specific $P$ value in the legend reflects the information reported by GraphPad Prism Version 9.4.0.

found that soluble tau aggregate uptake but not exposure to KRT8 or tau monomers triggered senescence-like changes in HBEC, including increased cell ($p < 0.05$) and nuclear size ($p = 0.025$, Fig. 3A–C), and increased expression of cell cycle arrest mediators Cdkn2a (p16, $p < 0.01$, Fig. 3D), Cdkn1a (p21, $p < 0.001$, Fig. 3E) and Tp53 ($p < 0.05$, Fig. 3F). Induction of cellular senescence is associated with the acquisition of a senescence-associated secretory phenotype (SASP)[27], which is prominent in endothelial cells undergoing senescence and has deleterious consequences for endothelial cell function and overall phenotype[48]. Treatment with soluble tau aggregates but not KRT8 potently induced SASP, including upregulation of interleukin 6 (IL-6, $p < 0.0001$, Fig. 3G), interleukin 1β (IL-1β, $p < 0.05$, Fig. 3H), TNF

($p < 0.001$, Fig. 3I), MCP-1 ($p < 0.05$, Fig. 3J), and PAI-1 ($p < 0.05$, Fig. 3K). These data indicate that soluble tau aggregate uptake triggers cellular senescence in primary human brain microvascular endothelial cells. Similar to the impact of tau monomers on microtubule density (Fig. 2C, D), we found that tau monomer exposure increased levels of components of the SASP (IL-6, IL-1β, and MCP-1, Fig. 3G, H and J) but had no impact on levels of cell cycle arrest markers (Fig. 3D–F), suggesting that inflammatory senescence-associated responses precede cell cycle arrest in human brain microvascular endothelial cells. Like changes in microtubule stability, increased expression of SASP in HBEC treated with tau monomers likely arise from the spontaneous oligomerization of monomeric tau during incubation (Supplementary Fig. 2A, B).

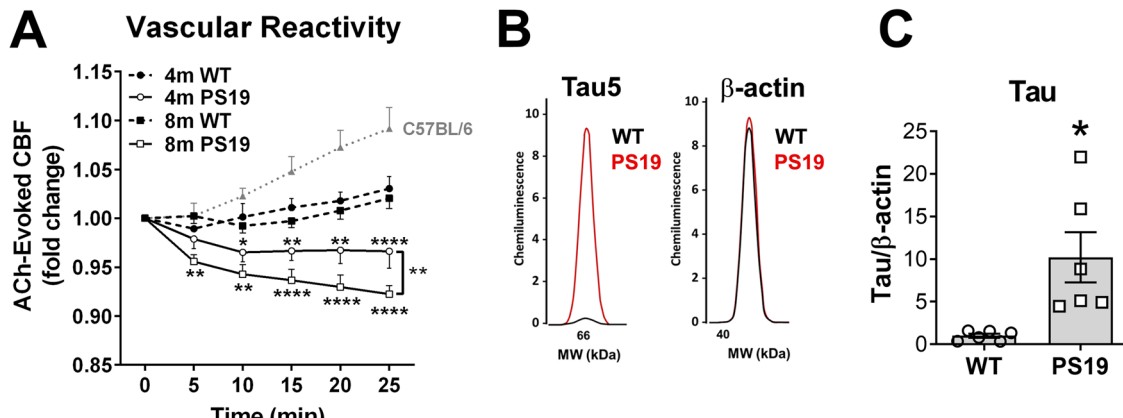

**Fig. 4 | Soluble tau aggregate accumulation in brain microvasculature of P301S(PS19) mice is associated with profound deficits in endothelium-dependent vasodilation. A** Impaired endothelium-dependent cerebral blood flow (CBF) responses in male and female P301S(PS19) mice in response to topical acetylcholine (ACh) stimulation as compared to WT animals worsen with age $(F(3,125) = 53.04$, ANOVA, $p < 0.0001$. * indicates Tukey's $q(126) = 3.80$, ** indicates Tukey's $q(126) = 4.66$, $p < 0.007$; **** indicates Tukey's $q(126) = 6.16$, $p < 0.0001$). The bracket highlights significant worsening of endothelial dysfunction with age in P301S(PS19) mice, ** Tukey's $q(126) = 4.61$, WT 4 months, $n = 6$; PS19 4 months, $n = 6$; WT 8 months, $n = 7$; PS19 8 months, $n = 6$ mice. Four-month-old animals were males. Eight-month-old animals were males and females. Data are means ± SEM, plotted against ACh-induced vasodilation in C57BL/6 J mice, that serve as reference and were not included in the analysis. **B**, **C** Accumulation of tau in brain microvasculature isolated from 8-month-old P301S(PS19) mice. **B** Representative electropherograms from capillary electrophoresis immunoassays showing increased tau in brain vasculature isolated from 8-month-old, male and female P301S(PS19) mice. **C** Quantitative analyses of data in (**C**) (Two-sided Student's t-test $t(5.064) = 3.116$, *, $p = 0.026$. Control, $n = 6$; PS19, $n = 6$ mice). Data are representative images and means ± SEM. For post-hoc analyses, lack of a specific P value in the legend reflects the information reported by GraphPad Prism Version 9.4.0.

## Accumulation of soluble tau aggregates in brain microvasculature is associated with impaired endothelium-dependent vasodilation in the P301S(PS19) mouse model of tauopathy

Pathogenic soluble tau aggregates accumulate in cortical capillaries and arterioles of AD and PSP brain[18] and in P301S(PS19) mice (Fig. 1A) in association with brain microvascular endothelial cells[18] (Fig. 1A). To define the impact of microvascular tau on brain microvascular function and better understand mechanisms of AD cerebrovascular damage, we measured endothelium-dependent vasomotion P301S (PS19) mice using laser Doppler flowmetry. Endothelium-dependent vasodilation, measured as cerebral blood flow (CBF) responses to topical acetylcholine (ACh) application, was profoundly impaired in both 4- and 8-month-old P301S(PS19) mice as compared to age-matched, wild-type (WT) littermates ($p < 0.0001$, Fig. 4A). Reduced or absent endothelial release of NO in conditions of severe endothelial dysfunction can lead to paradoxical vasoconstriction and vasospasm in response to ACh stimulation[49–55]. ACh stimulation of cerebral vasculature in P301S(PS19) mice led to paradoxical vasoconstriction measured as decreased CBF as compared to WT littermate mice (Fig. 4A). Of note, endothelium-dependent vasodilation to ACh stimulation in 4-month- and 8-month-old P301S(PS19) WT littermate control mice in an inbred C57BL/6JxC3H background were much lower than those in ACh-stimulated C57BL/6 J mice (Fig. 4A), suggesting that brain vascular endothelium of mice in the C57BL/6JxC3H background is less responsive to ACh stimulation than that of C57BL/6 J animals. Tauopathy arising from overexpression of human tau, however, profoundly impaired endothelial responses in P301S(PS19) mice, such that ACh stimulation resulted in paradoxical vasoconstriction, likely as a result of the direct action of ACh on vascular smooth muscle cells in the near absence of endothelium-dependent contributions. To further define the mechanisms by which overexpression of mutant human tau impairs vasodilation, we acutely isolated brain microvessels from P301S(PS19) and WT littermate mice and measured NO production using DAF-FM. Brain microvessels isolated from P301S(PS19) mice produced significantly less NO as compared to microvessels isolated from WT littermate control mice ($p = 0.024$, Supplementary Fig. 3A, B). Together with our studies of Fig. 4A, these data indicate that mutant human tau

overexpression reduces microvascular NO release and impairs endothelium-dependent vasodilation in P301S(PS19) mice modeling tauopathy.

In addition to endothelial cells, our earlier studies in AD and PSP brain also identified smooth muscle as a site of accumulation of soluble pathogenic tau aggregates[18]. Thus, to define the potential impact of pathogenic soluble tau aggregates on vascular smooth muscle cells we assessed the response of cortical microvasculature to sodium nitroprusside (SNP[56]), a nitric oxide donor. While brain vascular endothelium responses to ACh were severely decreased in P301S(PS19) mice (Fig. 4A), responses to SNP were comparable to those of WT littermates ($p = 0.63$, Supplementary Fig. 4) in P301S(PS19) mice, indicating that brain vascular smooth muscle function is unaffected by high levels of mutant human tau expression. Therefore, our data suggest that mutant human tau expression abrogates endothelium-dependent vasodilation by decreasing NO production in brain microvasculature of P301S(PS19) mice without detriment to brain vascular smooth muscle cell function.

Consistent with our observations in human AD brain[18], and the observed accumulation of soluble oligomeric tau aggregates in cortical microvasculature of 8-month-old P301S(PS19) mice (Fig. 1A) we found significantly higher levels of total tau in isolated brain microvasculature ($p = 0.026$, Fig. 4B, C) and in the cortex (Supplementary Fig. 5A–C) as compared to WT controls[18]. The accumulation of pathogenic soluble tau in brain microvasculature of P301S(PS19) mice, however, was not associated with a detectable loss of blood-brain barrier function or an increase in microbleeds ($p = 0.93$, Supplementary Fig. 5D, E). These data indicate that the accumulation of soluble tau aggregates in brain microvasculature is associated with severe endothelial dysfunction but may not impact BBB integrity in P301S(PS19) mice modeling tauopathy.

## Decreased microtubule stability, impaired eNOS activation, and increased markers of cellular senescence in brain microvasculature of P301S(PS19) mice

Consistent with our observations in primary human brain microvascular endothelial cells (Figs. 1–3), accumulation of soluble tau aggregates in brain microvasculature of 8-month-old P301S(PS19)

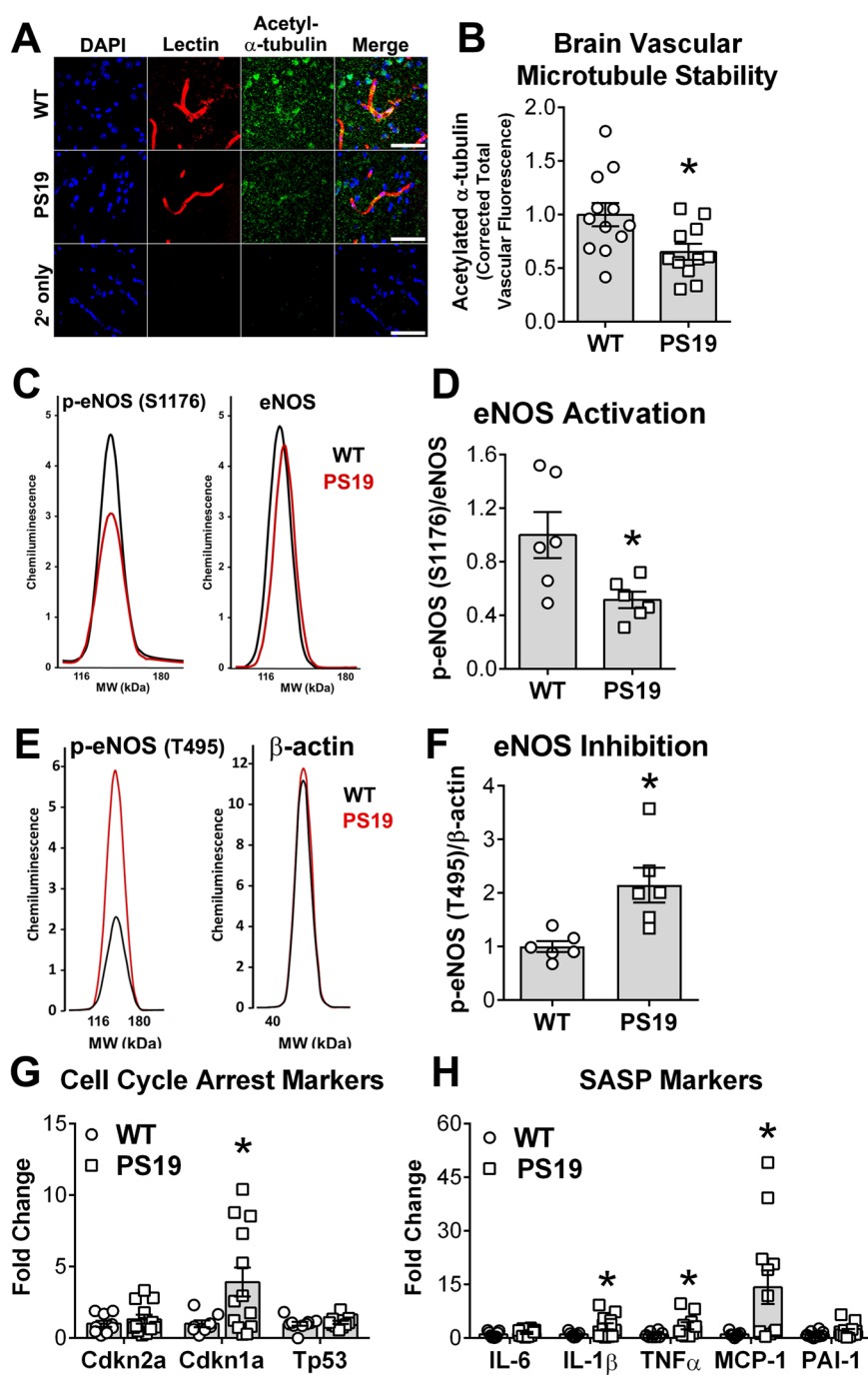

mice (Fig. 1A) was associated with decreased microtubule stability ($p = 0.018$, Fig. 5A, B), blunted eNOS activation ($p = 0.036$, Fig. 5C, D), increased eNOS inhibition ($p = 0.0152$, Fig. 5E, F), and increased markers of cellular senescence, including expression of Cdkn1a (p21, $p = 0.015$, Fig. 5G) and components of the SASP: IL-1β ($p = 0.027$), TNFα ($p = 0.043$), and MCP-1 ($p = 0.018$, Fig. 5H). The composition of SASP in microvasculature purified from brains of P301S(PS19) mice was different from that in primary HBEC exposed to pathogenic soluble tau in vitro, likely due to profound differences in environment make-up in vivo of brain microvasculature vs. in vitro in monotypic cell culture conditions[57], and potentially species-specific differences (human endothelial cells in vitro, mouse endothelial cells in vivo). Taken together, our data indicate that destabilization of microtubules, impaired eNOS activation, and increased expression of senescence markers in brain microvasculature may explain severe endothelial cell

dysfunction in P301S(PS19) mice (Fig. 4A) with high levels of microvascular pathogenic soluble tau deposition (Fig. 1A).

**Removal of soluble tau aggregates with immunotherapy reduces levels of tau in brain microvasculature and ameliorates brain microvascular endothelial cell dysfunction in P301S(PS19) mice**
Our in vitro studies showed profound brain microvascular endothelial cell deficits (Figs. 1–3) in association with tau microvascular accumulation similar to that in AD and PSP brain[18] in P301S(PS19) mice expressing aggregation-prone human tau (Fig. 1A). These observations suggested that brain microvascular dysfunction (Fig. 4A) in P301S(PS19) mice may arise from pathogenic tau uptake by brain microvascular endothelial cells (Figs. 1–3). Tau oligomer-specific monoclonal antibody (TOMA[24,58,59]) recognizes soluble tau aggregates but does not recognize soluble, functional forms of tau or tau

**Fig. 5 | Soluble aggregated tau induces microtubule instability, increases markers of cellular senescence, and blocks eNOS activation in brain microvasculature in 8-month-old male and female P301S(PS19) mice modeling tauopathy. A**, **B** Decreased levels of acetylated tubulin in microvasculature of P301S(PS19) mice. **A** Representative immunofluorescent images of cortical brain sections from WT control and P301S(PS19) mice stained with DAPI (blue) and lectin (red) and immunostained for acetyl-α-tubulin (green); **B** Quantitation of acetyl-α-tubulin immunoreactive signal co-localized with lectin-reactive brain microvasculature (Unpaired, two-sided Student's t-test, t(21) = 2.582, *, *p* = 0.018. Control, *n* = 12; PS19, *n* = 11 mice). **C**, **D** Decreased eNOS activation in microvasculature isolated from P301S(PS19) mice. **C** Representative electropherograms from capillary electrophoresis immunoassays of phospho-eNOS (S1176) and total eNOS in microvasculature purified from brains of WT and P301S(PS19) mice. **D** Quantitative analyses of phospho-eNOS (S1176) normalized to total eNOS levels in isolated brain vasculature of P301S(PS19) mice (Unpaired, two-sided Student's t-test, t(6.248) =2.662, *p* = 0.036. *n* = 6 mice per group). **E**, **F** Increased inhibition of eNOS in microvasculature isolated from P301S(PS19) mice. **E** Representative electropherograms from capillary electrophoresis immunoassays of phospho-eNOS (T495) and β-actin in microvasculature isolated from brains of WT control and P301S(PS19) mice. **F** Quantitative analyses of data in E (Student's t test with Welch's correction, t(5.966) = 3.370, *, *p* = 0.0152, *n* = 6 mice per group). **G**, **H** Increased markers of senescence in microvasculature isolated from P301S(PS19) mice.

**G** mRNA abundance for cell cycle arrest mediators cyclin dependent kinase inhibitor 2 A (Cdkn2a, p16, unpaired, two-sided Student's t test with Welch's correction, t(19) = 0.77, *p* = 0.45. WT control, *n* = 9; PS19, *n* = 12), cyclin dependent kinase inhibitor 1 A (Cdkn1a, p21, unpaired, two-sided Student's t test with Welch's correction (t(13.3) = 2.794, *, *p* = 0.015. WT control, *n* = 8; PS19, *n* = 13) and tumor protein 53 (Tp53, p53, unpaired, two-sided Student's t test (t(19) = 0.5973, *p* = 0.56. WT control, *n* = 10; PS19, *n* = 11 mice); **H** Senescence-associated secretory phenotype components interleukin 6 (IL-6, unpaired two-sided Student's t test, t(17)=1.023, *p* = 0.32. WT control, *n* = 9; PS19, *n* = 10 mice), interleukin 1β (IL-1β, unpaired, two-sided Student's t test with Welch's correction, t(12.87) = 2.497, *p* = 0.027. WT control, *n* = 7; PS19, *n* = 12 mice), tumor necrosis factor α (TNFα, unpaired, two-sided Student's t test with Welch's correction, t(10.46) = 2.309, *, *p* = 0.043. WT control, *n* = 8; PS19, *n* = 10 mice), monocyte chemoattractant protein-1 (MCP-1, unpaired, two-sided Student's t test with Welch's correction, t(11.07) = 2.794, *, *p* = 0.018. WT control, *n* = 7; PS19, *n* = 12 mice), and plasminogen-activator inhibitor-1 (PAI-1, unpaired, two-sided Student's t test with Welch's correction, t(15.53) = 1.701, *p* = 0.11. WT control, *n* = 10; PS19, *n* = 12 mice), measured with quantitative real-time PCR (qRT-PCR) in isolated brain vasculature of WT and P301S(PS19) mice. Data are representative images and electropherograms, and means ± SEM. For post-hoc analyses, the lack of a specific *P* value in the legend reflects the information reported by GraphPad Prism Version 9.4.0.

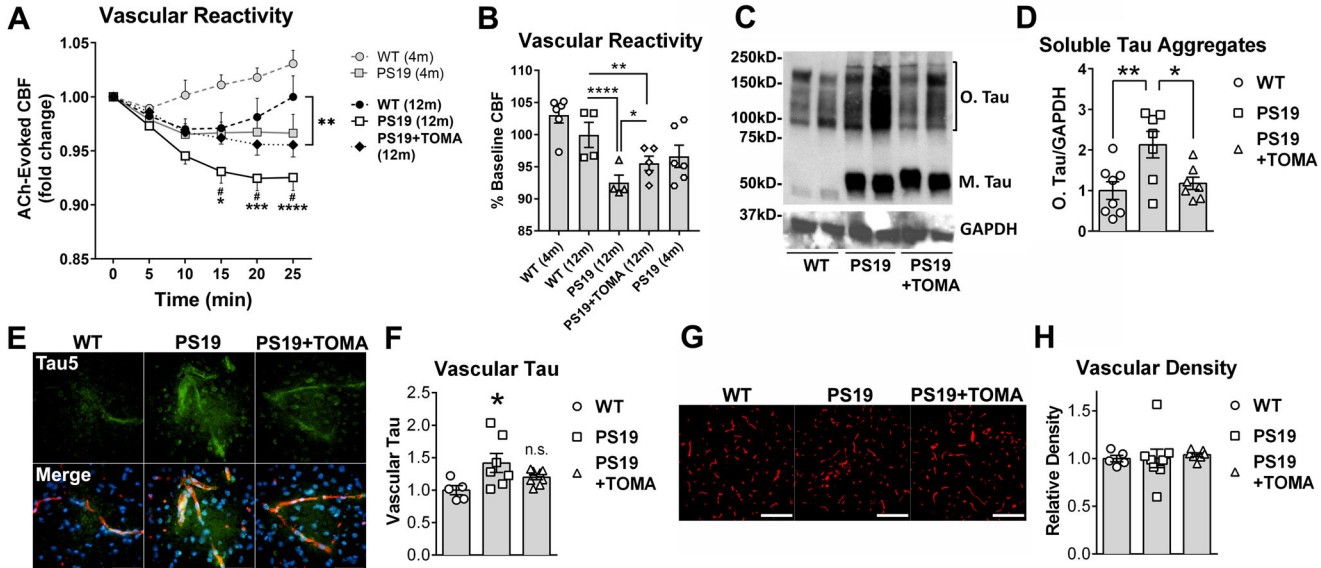

**Fig. 6 | Removal of soluble tau aggregates from the brain ameliorates brain microvascular deficits in 12-month-old male and female P301S(PS19) mice. A**, **B** Treatment with TOMA ameliorates profound deficits in endothelium-dependent vascular responses in somatosensory cortex of P301S(PS19) mice, restoring vascular responses to levels comparable to those of 4-month-old animals. **A** ACh-stimulated vascular responses; **B** Quantitative analyses of data at 25 min in A. Interaction of time and treatment group by 2WRM ANOVA, F(10,50) = 4.91, *p* < 0.0001; difference from WT by Holm-Sidak's *post hoc* indicated as *, t(60) = 2.86, *p* = 0.02; **, t(60) =3.33, *p* = 0.003; ***, t(60) = 4.04, *p* = 0.0005; ****, t(60) = 5.31, *p* < 0.0001; # indicates difference from PS19 + TOMA, for all #s, t(60)>2.34, *p* < 0.046 by Holm-Sidak's post hoc. WT, *n* = 4; PS19, *n* = 4; PS19 + TOMA, *n* = 5 mice. Data from 4-month-old WT and P301S(PS19) groups (Fig. 1) serve as reference and were not included in the analysis. **C**, **D** Treatment with TOMA reduces levels of soluble tau aggregates (tau oligomers) in brains of P301S(PS19) mice. **C** Representative immunoblots of cortical lysates of P301S(PS19) mice treated with isotype-matched IgG (PS19) or TOMA (PS19 + TOMA); D, Quantitative analyses of data in (C) (F (2, 19) = 6.807, ANOVA,

*p* = 0.0081. WT vs. PS19, **, *p* < 0.01; PS19 vs PS19 + TOMA, *, *p* < 0.05 by Tukey's *post hoc* test. WT, *n* = 8; PS19, *n* = 7; PS19 + TOMA, *n* = 7 mice). **E**, **F** TOMA treatment reduces levels of vascular tau in brains of P301S(PS19) mice. **E** Representative images (100X) of tau immunoreactivity (green) associated with lectin-reactive cortical brain vasculature (red), counterstained with DAPI (blue) staining. Scale bar is 50 μm. **F** Quantitative analyses of data in (**E**) (H = 6.839, Kruskal–Wallis test, *, *p* = 0.026. WT vs. PS19, *p* < 0.05 from Dunn's. n.s., no significant difference between experimental means for WT vs PS19 + TOMA. WT, *n* = 5; PS19, *n* = 7; PS9 + TOMA, *n* = 8 mice per group). **G**, **H** Cortical brain vascular density in P301S(PS19) mice is unchanged by tauopathy or by TOMA immunotherapy. **G** Representative images (40X) of lectin-reactive cortical brain vasculature (red) of P301S(PS19) animals treated with TOMA (PS19 + TOMA) or isotype-matched IgG (PS19). Scale bar is 250 μm. **H** Quantitative analyses of data in (**G**) (H = 1.787, Kruskal–Wallis, *p* = 0.426. WT, *n* = 5; PS19, *n* = 8; PS19, *n* = 7 mice). Data are representative images and means ± SEM. For post-hoc analyses, a lack of a specific P value in the legend reflects the information reported by GraphPad Prism Version 9.4.0.

fibrils. To define whether pathogenic soluble aggregated tau (as opposed to other forms of tau) mediates brain microvascular dysfunction we measured endothelium-dependent brain microvascular responses in P301S(PS19) mice that were treated with TOMA immunotherapy or with same-isotype IgGs. Effective doses of TOMA and

duration of treatment have been previously defined[24,58,59]. A single course of TOMA treatment administered in the symptomatic phase of tauopathy (8 weeks, starting at 5.5 months of age, after the onset of microvascular deficits, Fig. 4A), was sufficient to ameliorate significant deficits in endothelium-dependent vascular responses to topical ACh

application in brains of 12-month-old P301S(PS19) mice ($p = 0.046$, Fig. 6A, B), restoring brain vascular reactivity in TOMA-treated P301S(PS19) mice at advanced stages of tauopathy to levels similar to those in the early phases of tauopathy at 4 months of age (Fig. 6A, B). TOMA treatment significantly decreased the amount of soluble tau aggregates in the cortex of 12-month-old P301S(PS19) mice as compared to untreated controls ($p < 0.05$, Fig. 6C, D). In agreement with these observations, we found that cortical microvascular tau deposits in this cohort were increased in P301S(PS19) non-specific IgG-treated animals ($p = 0.026$, Fig. 6E, F), but not in TOMA-treated P301S(PS19) mice, as compared to WT controls (Fig. 6E, F). In contrast, fibrillar tau abundance in P301S(PS19) mice was not affected by TOMA treatment (Supplementary Fig. 6A, B). These data are consistent with previous reports of TOMA's specificity for soluble tau aggregates[59]. Prior studies have demonstrated that TOMA and other anti-tau antibodies transiently elevate plasma tau levels one hour after administration, for at least one week[59,60]. In our studies, plasma tau was elevated in P301S(PS19) mice ($p < 0.05$) compared to WT and P301S(PS19) treated with TOMA had similar levels to IgG treated P301S(PS19) mice (Supplementary Fig. 6C). Our plasma tau data may reflect the advanced state of tau pathology in P301S(PS19) mice at 12 months of age and the large time gap between the last TOMA dose (7.5 months of age) and plasma collection (12 months of age), that preclude detection of transient changes in plasma tau. Taken together, these data show that the removal of soluble pathogenic tau aggregates from the brain by a single course of TOMA treatment effectively restores endothelium-dependent vascular function in P301S(PS19) at advanced stages of tauopathy (Fig. 6A, B). Our data indicate that soluble aggregated tau has causal role in the etiology of microvascular dysfunction in P301S(PS19) mice. A limitation, however, is that TOMA is expected to reduce levels of soluble tau aggregates in multiple brain cell types, and potentially in the periphery. Thus, whether the restoration of microvascular function by TOMA is driven by the removal of soluble tau aggregates from microvascular endothelial cells alone, or in combination with the removal of pathogenic tau from other brain cell types or potentially from cell types of the periphery cannot be ascertained from our data.

Brain vascular density was unchanged in 12-month-old P301S(PS19) mice as compared to aged matched controls ($p = 0.43$, Fig. 6G, H), ruling out a potential impact of decreased vascular density on functional outcomes (Fig. 6A, B). Taken together, these data indicate that soluble pathogenic tau aggregates contribute significantly to brain microvascular dysfunction in a model of tauopathy.

## Discussion

Brain microvascular dysfunction plays an important role in the etiology and progression of AD[19,20] and related dementias associated with tauopathy[21]. Prior studies showed brain microvasculature abnormalities, including decreased vessel diameter, obstructed blood flow[61], and BBB breakdown[22] in mouse models of tauopathy but whether pathogenic tau has a role in brain microvascular dysfunction of AD remains unclear.

Prefibrillar soluble tau aggregates (tau oligomers) contribute significantly to the pathogenesis of AD and other dementias[5–7,62]. Neuronally-originated soluble tau aggregates, including oligomers, are released in the interstitial space[10,11], where they can reach nearby brain vascular elements. Consistent with this notion, we previously showed that pathogenic soluble tau oligomers accumulate in brain microvasculature of AD and other tauopathies, specifically in endothelial and smooth muscle cells[18]. The present studies indicate that accumulation of pathogenic soluble tau aggregates in AD brain microvasculature is recapitulated in the P301S(PS19) mouse model of tauopathy and is associated with profound deficits in brain microvascular function that can be negated by removal of soluble tau aggregates with immunotherapy. We show that, similar to neuron-to-neuron transmission of

misfolded tau[10,12,14], extracellular soluble tau oligomers can enter human brain microvascular endothelial cells, and that like for neurons this entry is mediated by binding cell-surface HSPG. Taken together our observations suggest that soluble tau aggregate entry to endothelial cells mediates the accumulation of pathogenic tau in AD and in P301S(PS19) mice, and are consistent with prior studies that documented peripheral clearance of tau following anti-tau antibody administration in tauopathy patients and in mice modeling tauopathy, and showed that tau isoforms can cross the BBB[23,24,60].

Trans-neuronal transmission of extracellular tau induces endogenous tau phosphorylation and aggregation after cellular uptake[10,12,16] with consequent destabilization of the microtubule cytoskeleton[1,3]. Although tau protein is primarily considered an axonal marker, in agreement with prior reports in the present studies we show that tau is also expressed in brain vascular endothelial cells[25,26]. Like trans-neuronal transmission of tau, entry of extracellular tau aggregates into human brain microvascular endothelial cells promoted endogenous native tau phosphorylation and subsequent destabilization of the microtubule cytoskeleton. These findings have important pathophysiological relevance since the microtubule cytoskeleton plays a critical role in various signaling pathways in brain vascular endothelial cells, including but not limited to mechanotransduction of shear stress, BBB maintenance, and transport eNOS to membrane compartments, which enables eNOS activation[40,46] by binding to HSP90[40] and by phosphorylation at Ser1176/1177[39]. Consistent with this knowledge, we found that eNOS translocation to membrane compartments of human brain endothelial cells was significantly reduced, causing a significant decrease in phosphorylation-mediated eNOS activation, a significant increase in phosphorylation-mediated eNOS deactivation, and a significant decrease in NO release. Taken together, these data support a model where pathogenic tau destabilizes the microtubule cytoskeleton in human brain endothelial cells, inhibiting eNOS activation and NO release by disrupting transport of eNOS to membrane compartments.

Conflicting data have been reported for studies that examined the subcellular location of active forms of eNOS. Hecker et al. reported that there are two major pools of activated eNOS: one associated with the cell membrane and one associated with the endoplasmic reticulum[42]. Govers et al.[44] also reported that eNOS activation is associated with its presence at cell-cell contacts; this would explain why cultures of endothelial cells need to be close to confluency to observe substantial eNOS activation. It has also been reported that while eNOS becomes unbound from caveolin for activation, the enzyme remains at the cell membrane, is phosphorylated at S1177, and interacts with calmodulin and Hsp90 to become active[43,45]. Our data shows that treatment with pathogenic oligomeric tau decreases levels of eNOS at the cell membrane in HBEC stimulated with acetylcholine. We also found an apparent reduction in eNOS localized to the endoplasmic reticulum, although this difference was not significant. The impact of pathogenic oligomeric tau on eNOS localization in HBEC is consistent with the observed decrease in NO production, reduction in active, S1177-phosphorylated eNOS, and increase in eNOS phosphorylated at its inhibitory site, T485, arising from pathogenic tau exposure. Other reports however, have suggested that activation of eNOS is associated with its translocation from the membrane to the cytosol[63,64]. It is possible that these apparent conflicts in the literature may reflect endothelial cell subtype-specific and potentially also cell state-specific differences in localization of eNOS-interacting proteins required for its activation (e.g. localization of pools of various interaction partners of the enzyme) or in other factors present in different types of endothelial cells (e.g. cardiac endothelial cells vs. brain endothelial cells, activated vs. quiescent endothelial cells).

Endothelial dysfunction, manifested as impaired endothelium-dependent NO-mediated vasodilation, is a critical mechanism of CBF

dysregulation both in aging[65,66] and in age-related neurodegenerative diseases[67,68] and is causally linked to cognitive decline[66]. Because of its central role in the regulation of vascular tone, reduced eNOS activation associated with a significant reduction in NO production are expected to impair endothelial function and thus drive CBF changes associated with aging and age-associated disease, including but not limited to impaired endothelium-dependent vasodilation, that we showed is prominent in P301S(PS19) mice. While our data shows that pathogenic tau has no significant impact on vascular smooth muscle function, the etiology of pathogenic tau-induced microvascular dysfunction could be multifaceted and involve multiple brain microvascular cell types that compose the neurovascular unit (including possibly astrocytes and pericytes) and may thus not be limited to tau impact on endothelium. Future studies will clarify this distinction, and also define the contribution of pathogenic tau-induced damage in astrocytes and pericytes to brain microvascular dysfunction in AD and other tauopathies.

That partial removal of pathogenic soluble aggregated tau from brains of P301S(PS19) mice using immunotherapy can negate, at least in part, the impact of pathogenic tau species on the microvascular response to acetylcholine in this mouse model indicates that pathogenic soluble tau aggregates, but not fibrillar or non-pathogenic monomeric forms of tau, are causally involved in microvascular dysfunction of P301S(PS19) mice. These data are consistent with a wealth of data supporting the notion that soluble aggregated forms of tau are key mediators of tau toxicity[6,69,70]. Importantly, these data have immediate translational implications since TOMA is being developed for clinical trials (Dr. Kayed, personal communication).

While previous research has focused on the pathogenic role of Aβ in impairment of eNOS-dependent vasodilation[67,68], our studies provide evidence for mechanisms of pathogenic tau-induced endothelium-mediated cerebromicrovascular dysfunction in AD. Our data suggest that entry of soluble misfolded tau followed by destabilization of the microtubule cytoskeleton in brain microvascular endothelial cells may underlie defects in endothelium-dependent vasodilation in tauopathies. Interestingly, eNOS ablation was shown to increase levels of p25 kinase, which phosphorylates tau, leading to cognitive dysfunction[71]. Furthermore, it was recently reported that loss of eNOS activity in APP/PS1/eNOS$^{-/-}$ mice exacerbates tau phosphorylation and cognitive decline[71]. Thus, entry of pathogenic tau into brain vascular endothelial cells may trigger a vicious cycle involving decreased eNOS activity, aberrant tau phosphorylation, and microtubule destabilization.

Large studies in human cohorts have suggested that vascular defects precede both amyloid-beta (Aβ) and phosphorylated tau accumulation[20]. While our studies do not address this aspect of AD pathogenesis, our data suggest that, similar to Aβ, the accumulation of pathogenic oligomeric forms of tau exacerbate vascular dysfunction, contributing together with Aβ to establish a feed-forward loop of vascular damage with Aβ and pathogenic oligomeric tau accumulation during the progression of AD. While we did not observe changes in blood-brain barrier permeability as measured by fibrinogen extravasation in P301S(PS19) mice at the ages tested, our studies show significant damage to endothelial cell function caused by pathogenic oligomeric forms of tau in P301S(PS19) mice.

Activation of cellular senescence in vascular endothelial cells underlies age-associated vascular dysfunction and initiates vascular disease[72]. Remarkably, we found that entry of pathogenic tau robustly induced cellular senescence/SASP in primary human brain microvascular endothelial cells, and that overexpression of pathogenic tau in vivo was sufficient to increase markers of cellular senescence in brain microvasculature of P301S(PS19) mice. While entry of soluble recombinant pathogenic tau into human brain microvascular endothelial cells caused upregulation of practically all senescence-associated markers of cell cycle arrest and SASP that were examined, a smaller subset of cell cycle arrest and SASP markers were upregulated in brain microvascular fractions isolated from P301S(PS19) mice. It has been shown that the SASP composition and strength varies substantially among cell types[57] and also within a given cell type depending on multiple factors, including prominently upstream signals from its milieu[29,73–75]. Cultured human brain endothelial cells were exposed to a single, relatively homogeneous dose of purified soluble tau aggregates in defined cell culture conditions in vitro. In contrast, mouse brain endothelial cells undergoing senescence in vivo were embedded in the diverse multicellular niche of the brain microvasculature where, in addition to pathogenic soluble tau aggregates, they were continuously exposed to other cell types and to a host of substances present in the brain interstitial fluid, including those specific to the disease state in P301S(PS19) mice undergoing tauopathy. Thus, the differences in profile of inflammatory factors secreted during senescence by human brain endothelial cells in vitro as compared to that of mouse brain endothelial cells undergoing senescence in vivo may arise from numerous differences in the environmental milieu that these cells were exposed to while undergoing pathogenic tau-induced senescence.

Of note, increased eNOS activity can block endothelial cell senescence[30]. Therefore, pathogenic tau-induced endothelial cell senescence may be further exacerbated by reduced eNOS activation, and it is possible that both reduced eNOS activation and cellular senescence[47] are triggered by pathogenic tau-induced destabilization of the microtubule cytoskeleton. Previous reports have shown that microtubule destabilization induces inflammation[76–78] and that low doses of the microtubule stabilizer, paclitaxel, can reduce levels of inflammation[79]. Our finding that soluble tau aggregates spontaneously generated in tau monomer preparations trigger microtubule destabilization and activate inflammatory responses but do not impact eNOS activity nor trigger cell cycle arrest during incubation with primary human brain microvascular endothelial cells provide further support for a model where microtubule cytoskeleton destabilization and activation of inflammatory gene networks precedes eNOS deactivation and cell cycle arrest in human brain vascular endothelial cells. This model suggests that soluble tau aggregate uptake may trigger a cascade of events where microtubule destabilization and inflammation precede eNOS inhibition and cell cycle arrest, suggesting that removal of soluble tau aggregates with immunotherapy early in this process may effectively block potentially irreversible phenotypic changes in endothelial cells, preserving brain vascular nitric oxide bioavailability and blocking pathogenic tau-induced endothelial cell senescence.

In conclusion, the present studies provide evidence for entry of soluble tau oligomers to brain endothelial cells and specify its impact on brain microvascular function. Our data suggest that the accumulation of soluble pathogenic tau in AD microvasculature[18] and in models of amyloidopathy[18] and tauopathy may be explained by the entry of soluble pathogenic tau into brain microvascular endothelial cells. We propose that microtubule destabilization following pathogenic soluble tau entry into endothelial cells impairs eNOS activation, leading to severe cerebrovascular dysfunction. Furthermore, induction of endothelial cell senescence by pathogenic tau entry is expected to change endothelial cell phenotype and trigger further endothelial dysfunction, exacerbating brain microvascular injury. Taken together, our observations broaden the impact of soluble pathogenic tau released extracellularly and suggest that in addition to mediating trans-neuronal propagation of tau, pathogenic tau may also propagate tau pathology to brain microvascular endothelial cells, thereby contributing to the genesis of vascular dysfunction in Alzheimer's disease and other tauopathies.

## Methods

### Cell culture and treatments

Primary human brain microvascular endothelial cells (HBEC) purchased from Cell Biologics (H-6023, Chicago, IL, USA) were cultured on gelatin-coated plates with complete human endothelial cell medium (Cell Biologics H1168, Chicago, IL, USA) in a 5% $CO_2$ humidified incubator at 37 °C. All cells were plated at the same density prior to treatment. HBEC were treated with treated with 40 µg/mL tau oligomers (O. Tau), 40 µg/mL monomeric tau (M. tau), 40 µg/mL recombinant human cytokeratin-8 (KRT8, RayBiotech, Norcross, GA, USA), or vehicle (PBS) in 1 mL of complete media in a 35 mm plate for 42 h on a rocker (15 RPM) to simulate bidirectional low wall shear stress.

### Animals

All studies were performed under approval of the UTHSCSA Institutional Animal Care and Use Committee (Animal Welfare Assurance Number: A3345-01) and in compliance with the ARRIVE guidelines (Animal Research: Reporting In Vivo Experiments) for reporting animal experiments. Male and female P301S(PS19) mice expressing human tau carrying the P301S mutation in the C57BL/6JxC3H were generated in our laboratory from original breeders obtained from The Jackson Laboratory (Stock #008169). At 3 months of age total tau (but not hyperphosphorylated tau) increases in P301S(PS19) mice, together with synaptic loss and microgliosis in hippocampus, amygdala, cortex, brain stem, and spinal cord. By 6 months of age, total tau and hyperphosphorylated tau increase in the same brain regions accompanied by neurofibrillary tangles and increased astrogliosis. Neuron loss together with hippocampal and cerebral cortical atrophy begins at 9 months of age[80]. All mice were housed ≤5/cage and maintained on a 12 h light/12 h dark cycle at 75 ± 3°F and relative humidity ≥30%. Animals were euthanized by cervical dislocation under ketamine/xylazine anesthesia after assessment of vascular reactivity (see below) or isoflurane overdose if not used in vascular reactivity measures. Cardiopulmonary arrest was verified before the brain was dissected out, within 1-1.25 minutes after death. We have verified that our approach effectively circumvents vascular or other tissue artifacts in all our experiments using immunohistochemistry for vascular markers. No vascular or parenchymal abnormalities nor autofluorescence indicative of tissue damage, hemorrhage, and other injury in brain tissues of healthy wild-type non-transgenic groups were detected following cervical dislocation[67,81–83] including the studies reported in the present manuscript.

### Tau uptake

HBEC were exposed to 40 µg/mL recombinant human tau-441 (2N4R) with c-terminal His and V5 tags (Millipore Sigma SAE0076, St. Louis, MO, USA) or vehicle (PBS) for 2 h with or without and 0.5 mg/mL heparin sodium (Tocris, Minneapolis, MN, USA). Cells were fixed in 4% paraformaldehyde and immunocytochemistry was performed (see methods below).

### Preparation of soluble tau aggregates (tau oligomers)

Recombinant tau protein (tau-441 (2N4R) MW 45.9 kDa) was expressed and purified as described[5]. The tau pellet was treated with 8 M urea followed by overnight dialysis against 1X phosphate buffered saline (PBS), pH 7.4. Tau concentration was measured using bicinchoninic acid protein assay (Micro BCA kit, Pierce, Rockford, IL, USA) and diluted to 1-1.2 mg using 1X PBS. Aliquots of tau monomer in PBS were stored at −20 °C. Samples were diluted to the desired concentration using 1X PBS and incubated for 1 hour on an orbital shaker at room temperature. After shaking, the resulting tau oligomers were purified by fast protein liquid chromatography (FPLC, Superdex 200HR 10/30 column, Amersham Biosciences, Chicago, IL, USA). The oligomeric tau generated by this method is unphosphorylated.

### Atomic force microscopy (AFM)

Aliquots of 1 mM monomeric tau or oligomeric tau (soluble tau aggregates) preparations in PBS were diluted 200-fold to 5 µM in ultrapure $H_2O$. Media samples after incubation with HBEC containing either monomeric tau or KRT8 were diluted by 50-250X with ultrapure $H_2O$ depending on original sample concentration. Three microliters of the diluted preparations of purified tau or tau incubated with media were deposited on a freshly cleaved muscovite mica and left under a PCR hood at room temperature for 5 min. Then the mica was washed with $H_2O$ and gently dried in a stream of nitrogen. The mica was then mounted in a Multimode Nanoscope IIIa (Bruker Inc., Billerica, MA, USA) with an E scanner. The samples were imaged in a tapping mode in air, with TESP probes (Bruker Inc., Billerica, MA, USA), tuned to resonant frequency between 290 and 320 kHz, with amplitude between 70 mV and 100 mV, and a set point between 1.4 and 1.8 V. At least ten pairs of trace and retrace height-mode images of 1 µm² fields, each containing at least a hundred of protein particles, were acquired for each sample with a rate of 3 Hz. The raw images were flattened and plane-fitted with the Nanoscope III software. The analysis of particle length distribution was performed for at least 370 imaged particles for each sample with the SPIP software (v. 6.0.13; Scanning Probe Image Processor, Image Metrology, Lyngby, Denmark).

### Immunocytochemistry (ICC)

HBEC were rinsed twice with PBS and then fixed with 4% v/v paraformaldehyde for 20 minutes, permeabilized with 0.1% Triton X-100 in PBS for 15 minutes, and then blocked with 10% goat serum for 1 hour. HBEC were incubated with primary antibody (β-tubulin (Sigma Aldrich, T4026, St. Louis, MO, USA), α/β tubulin (Cell Signaling Technology 2148 S, Danvers, MA, USA), V5 antibody (Thermo Fisher Scientific R960-25, Waltham, MA, USA), Tau5 (Millipore, MAB361, Billerica, MA, USA)), and/or eNOS (AF950, R&D Systems, Minneapolis, MN, USA) overnight at 4 °C. Cells were incubated with the appropriate secondary antibody conjugated to Alexa Fluor 488 or 594 and counterstained with DAPI. Images were captured with a 40X objective using a Nikon Eclipse TE2000U inverted microscope (Nikon, Melville, NY, USA) using the NIS-Elements software 5.11.01 (Nikon, Melville, NY, USA). Additional antibody information is supplied in Supplementary Table 1.

### Measures of microtubule density

Cells immunostained with β-tubulin and DAPI (see ICC methods) were imaged at 20X magnification using confocal optical imaging (Operetta High-Content Imaging System) and analyzed using the Harmony High-Content Imaging data analysis platform (Perkin Elmer, Waltham, MA, USA). Microtubule density distributions along dissector lines systematically positioned distal to one of the nuclear and crossing the cytoplasm of randomly chosen cells for each of 6 fields in each sample, distributed evenly over the field were quantified by investigators blinded to treatment using ImageJ software (NIH, Bethesda, Maryland, USA). Microtubule density values were calculated as the area under the curve for values of β-tubulin immunoreactivity intensity as cross-sectional microtubule density. All density values per line were averaged for each cell and used to determine means for each group.

### eNOS Localization

HBEC were plated in a 96-well plate and treated with vehicle (PBS, control), 40 µg/mL monomeric tau (M. tau), or, 40 µg/mL tau oligomers (O. Tau) for 42 hours. HBECs were then treated with media only or 10 µM acetylcholine to induce eNOS translocation for 20 minutes. HBEC were fixed in PFA and immunostained with antibodies for eNOS and β-tubulin and stained with DAPI according to the methods described in the ICC methods, above. HBEC were imaged using the Zoe Fluorescent Cell Imager with software version 002.257.011215 (Bio-Rad Laboratories, Hercules, CA, USA). Images were analyzed using the plot profile function in Fiji ImageJ (NIH, Bethesda, MD, USA). eNOS

localization within the cell was separated into nuclear, perinuclear, cytoplasmic, and edge regions as determined by the plot profile (gray value versus distance in pixels from the center of the nucleus to the edge of the cell) and visualization of features within each individual cell. Six samples from two separate biological replicates were analyzed for each group using Prism 8 (GraphPad, San Diego, CA, USA).

## Nitric oxide (NO) production

HBEC were plated in a 96-well, optical bottom plate and treated with vehicle (PBS, control), 40 µg/mL monomeric tau (M. tau), or, 40 µg/mL tau oligomers (O. Tau) for 42 hours. HBEC were then loaded with the nitric oxide indicator DAF-FM diacetate (D-23844, ThermoFisher, Waltham, MA, USA) in media for 45 minutes at 37 °C. The media was then replaced with imaging media (100 mM HEPES, pH 7.4, 750 mM NaCl, 5 mM CaCl$_2$, 25 mM KCl, 5 mM MgCl$_2$, 1.9 µg/mL glucose, 1.9 mg/mL BSA) and incubated at 37 °C for 15 min to allow for the DAF-FM to be fully converted. Fluorescence from the DAF-FM was measured using the POLARstar Omega plate reader (ex485nm/em520nm, BMG LABTECH, Cary, NC, USA). Data shown are from 3 independent experiments.

To determine NO production in isolated brain microvasculature from WT and P301S(PS19) mice, microvasculature was isolated as described in the brain microvascular isolation methods section. Debris from the microvessels was removed by washing in cold media (DMEM (Sigma D1145) with 10% Cosmic calf serum HyClone, Logan, UT, USA), 50 µg/mL DNase I (Sigma Aldrich, St. Louis, MO, USA), 0.1 mM non-essential amino acids (Gibco, Grand Island, NY, USA), and 2 mM GlutaMax (Gibco)) and straining through a 70 µm cell strainer to remove cell debris. The filtered vessels were collected from the filter with more cold washing media and pelleted. Following centrifugation, pellets were resuspended in media (DMEM (Sigma D1145) with 20% Cosmic calf serum (HyClone), 0.1 mM non-essential amino acids (Gibco), 2 mM GlutaMax (Gibco), 25 mM HEPES, 1 mM sodium pyruvate, and 15 U/mL heparin) containing DAF-FM diacetate (D-23844, Thermo Fisher, Waltham, MA) and incubated for 40 min at 37 °C. During the final minutes of incubation 10 µg/mL Hoeschst 33342 nuclear dye was added to the media. Vessels were centrifuged at 37 °C for 10 minutes and resuspended in warm imaging media (100 mM HEPES, pH 7.4, 750 mM NaCl, 5 mM CaCl$_2$, 25 mM KCl, 5 mM MgCl$_2$, 1.9 µg/mL glucose, 1.9 mg/mL BSA). Microvessels were plated in a 96-well plate and imaged with the ImageXpress Confocal HT.ai system (Molecular Devices, San Jose,CA, USA). DAF-FM (green) fluorescence intensity in isolated microvessels was measured using ImageJ with the Fiji extension (NIH, Bethesda, MD, USA) and normalized to both vessel area and to vessel diameter. Hoeschst 33342 nuclear staining images were overlayed with DAF-FM images to unequivocally identify vascular elements where DAF-FM staining was quantified. Data are averages of fluorescence intensity/area/diameter representing the amount of NO produced in isolated brain microvessels of WT and P301S(PS19) mice. To account for differences in total numbers of vessels isolated from each animal, total fluorescence values were weighted so that each animal contributed equally to the average for each genotype group. Data are from 3 individual animals per genotype with ≥57 brain microvessels measured per animal with a total of 372 microvessels analyzed in PS19 animals and 436 microvessels in WT controls.

## Traditional Western blot

For levels of tau in HBEC, cellular protein lysates were separated using SDS-PAGE (Bio-Rad, 4–15% Mini-PROTEAN TGX, Hercules, CA, USA) and transferred onto Amersham Protran nitrocellulose (GE Healthcare, Chicago, IL, USA) and blocked with 5% BSA. The membrane was incubated in primary antibody overnight (Tau5 (Millipore, MAB361, Billerica, MA, USA) and β-actin (Cell Signaling Technology, 4970 S, Danvers, MA, USA)) and then incubated with LI-COR near-infrared fluorescent secondary antibodies for 1 h at room temperature. Western

blots were imaged using the LI-COR Odyssey and analyzed using the Image Studio Lite (version 5.2) software (LI-COR, Lincoln, NE, USA). Additional antibody information is supplied in Supplementary Table 1.

To quantify tau oligomers (soluble tau aggregates) by traditional Western blot, brain tissue lysates were generated from prefrontal cortex by homogenization in PBS containing protease inhibitor (cOmplete mini, Roche, Indianapolis, IN, USA) and 1 mM sodium orthovanadate using the Bullet Blender Storm 24 (Next Advance, Troy, NY, USA). Protein lysates were separated using SDS-PAGE (Bio-Rad, 7.5% Mini-PROTEAN TGX, Hercules, CA, USA) and transferred to nitrocellulose membranes (GE Healthcare, Chicago, IL, USA). Membranes were blocked with 10% non-fat milk and incubated in primary antibody diluted in 5% non-fat milk overnight (Tau5 (1:1000, Millipore, MAB361, Billerica, MA, USA) and GAPDH (Thermo Fisher Scientific #PA1-987, Waltham, MA, USA)). Membranes were incubated with appropriate HRP-conjugated secondary antibodies and incubated with ECL substrate (ThermoFisher, Waltham, MA, USA). Membranes were exposed to chemiluminescence film (GE Healthcare, Chicago, IL, USA) and developed. Films were analyzed using ImageJ software (NIH, Bethesda, Maryland, USA). Soluble tau aggregates (tau oligomers) were quantified as Tau5-positive bands between ~85–240 kD and density was normalized to GAPDH. The number of samples used in experiments of Fig. 6C, D precluded the use only one gel/blot for their analysis. Samples from different experimental groups, however, were randomized to each blot such that an equal number of each experimental group was ran in each of those gels/blots.

## Capillary electrophoresis immunoblots

HBEC were harvested with 1X cell lysis buffer (Cell Signaling Technology, Danvers, MA) with protease inhibitors (cOmplete mini, Roche, Indianapolis, IN, USA). Vascular homogenates were generated as described in the brain microvascular isolation methods section. Capillary electrophoresis immunoblots were performed using the ProteinSimple Wes system (San Jose, CA, USA). The following primary antibodies were used: phospho-eNOS S1176/1177 (BD Biosciences, #612392, San Jose, CA, USA), phospho-eNOS T495 (BD Biosciences, #612706, San Jose, CA, USA), eNOS (Millipore, 07-520, Billerica, MA), phospho-tau T231 (AT180, Thermo Scientific, MN1040, Waltham, MA, USA), β-actin (Cell Signaling Technology, 4970 S, Danvers, MA, USA), acetyl-α-tubulin (Santa Cruz Biotechnology, sc-23950, Dallas, TX, USA), and α-tubulin (Cell Signaling Technology, 2144 S, Danvers, MA, USA). Data was analyzed using the Compass for SW software (ProteinSimple, San Jose, CA, USA). Chemiluminescent signals were measured as area under the curve. Additional antibody information is supplied in Supplementary Table 1.

## Cell and nuclear size quantification

Following treatment (see Cell culture and treatment methods above), cells were incubated with 0.3 µg/mL Hoescht 33342 for 30 minutes and then imaged. Each condition was imaged in triplicate wells with 7 fields per well using the Harmony High-Content Imaging analysis platform (Perkin Elmer, Waltham, MA, USA). Cellular and nuclear sizes were determined using cell method A with the Harmony software.

## Quantitative real-time -PCR (qRT-PCR)

Total RNA from primary HBEC was isolated using the RNAqueous-4PCR kit according to the manufacturer's instructions (Life Technologies, Carlsbad, CA, USA), cDNA was generated from the total isolated mRNA using the SuperScript III First-Strand kit according to the instructions provided (Life Technologies Carlsbad, CA, USA), and quantitative RT-PCR was conducted using the Power SYBR Green master mix (Life Technologies Carlsbad, CA, USA). Total RNA from isolated P301S(PS19) mouse brain vasculature was extracted using the Qiagen AllPrep DNA/RNA/Protein Mini Kit (Germantown, MD, USA), cDNA was prepared using the Bio-Rad iScript cDNA synthesis kit, and

qRT-PCR reactions were completed using the Bio-Rad iTaq Universal SYBR Green Supermix (Bio-Rad, Hercules, CA, USA). Quantitative RT-PCR reactions were performed using the 7900HT Sequence Detection System (Applied Biosystems, Foster City, CA, USA). The CT for each sample was calculated using the SDS 2.3 software (Applied Biosystems Foster City, CA, USA). Relative mRNA expression levels were calculated using the comparative method ($2^{-\Delta\Delta CT}$) using GAPDH as a reference. Primers used for qRT-PCR are shown in Supplementary Table 2.

### Immunofluorescence in brain sections
Ten micrometer coronal cryosections from P301S(PS19) and WT brains were post-fixed in 4% paraformaldehyde and stained with antibodies for oligomeric tau (T22[6], total tau (Tau5, Millipore, MAB361, Billerica, MA, USA), paired helical filament-tau (phospho-tau, AT8, MN1020, Thermo Fisher Scientific, Waltham, MA, USA), CD31 (BD Biosciences, #553370, San Jose, CA, USA), fibrinogen (Dako, A0080, Santa Clara, CA, USA), and acetylated α-tubulin (Santa Cruz Biotechnology, sc-23950, Dallas, TX, USA), followed by the appropriate fluorescently-tagged secondary. DyLight® 594 Lycopersicon esculentum (tomato) lectin (Vector Laboratories, Burlingame, CA, USA) was used to highlight brain vasculature in cryosections. Slides were imaged with a laser scanning confocal microscope (Zeiss LSM 780, San Diego, CA, USA) using an argon laser. Stacks of confocal images for each channel were obtained separately at $z = 0.52\,\mu m$ using a 40X water immersion objective with a numerical aperture of 1.1. Z-stacks of confocal images were processed using ImageJ (NIH, Bethesda, MD, USA). Images were quantified using ImageJ (NIH, Bethesda, MD) by analyzing and averaging 10 μm z-stacks with images taken at 1 μm intervals. Regions of interest (ROIs) were identified for the vasculature (lectin) in each image and then the ROI for each image was transferred to the channel containing acetylated α-tubulin. Brain vascular acetylated α-tubulin fluorescence intensity was then measured based on the corrected total cell fluorescence formula utilizing ROIs containing the brain vasculature.

### Vascular reactivity
Cerebral blood flow (CBF) responses to acetylcholine, an endothelium-dependent vasodilator, were measured using laser Doppler flowmetry. Animals were anesthetized with a mixture of ketamine (85 mg/kg) and xylazine (15 mg/kg), were intubated and ventilated, and placed in a stereotaxic frame on a heating pad to maintain body temperature. Blood pressure (CODA Monitor, Kent Scientific, Torrington, CT, USA), blood oxygen saturation, heart rate, peripheral capillary oxygen saturation, and end-tidal $CO_2$ (PhysioSuite, Kent Scientific) were monitored throughout the procedure. The bone over the left somatosensory barrel cortex (0.5–1.5 mm posterior and 3–4.5 mm lateral from bregma) was removed, a silicone barrier was made around the craniotomy, aCSF was applied, and a laser Doppler probe (Transonic, Ithaca NY, USA) was positioned over the barrel cortex. After baseline CBF had been established, the existing aCSF was replaced with 15 μL of 10 μM acetylcholine (ACh) (Acros Organics, New Jersey, USA) in aCSF to elicit endothelium-dependent vasodilation. CBF was subsequently measured for 25 minutes and the ACh solution was replaced every 5 minutes. Data was collected with Lab Chart 8.0.7 software. Changes in CBF are expressed as fold change relative to baseline recordings in 5-minute increments after ACh application. Data were analyzed with a two-way (genotype x time), repeated measures ANOVA, followed by Sidak's multiple comparison post hoc test to compare group means at each time point.

### Vascular smooth muscle relaxation
Mice were anesthetized and prepared as described for vascular reactivity. A stable baseline cerebral blood flow was acquired using laser Doppler flowmetry with aCSF superfused over the cranial window. Subsequently, 20 μM sodium nitroprusside (SNP, Sigma-Aldrich, St. Louis, MO), a nitric oxide donor that acts on vascular smooth muscle to produce vasodilation, was continuously superfused for 10 minutes, and the change in cerebral blood flow was recorded and expressed as fold change from baseline.

### Tau oligomer-specific monoclonal antibody (TOMA) immunotherapy
WT and P301S(PS19) mice were treated with 60 μg of TOMA[59] or same-isotype IgG (Rockland Immunochemicals, Limerick, PA, USA) via tail vein injection, every 2 weeks starting after disease onset at 5.5 months of age and continuing for 8 weeks until 7.5 months of age. TOMA was developed and validated in the Kayed laboratory[59], where dose and regimen for TOMA immunotherapy were established[24,58,59,84], with no discernible side effects in P301L and Htau mice[24,59]. Animals were sacrificed and tissues were collected after brain microvascular reactivity was measured at 12.5 months of age. Plasma was collected and separated in plasma separator tubes (BD Biosciences, #365965, San Jose, CA, USA) at time of sacrifice.

### Measurement of plasma tau by ELISA
Plasma was diluted at a 1:20 ratio and tau was quantified using a human tau ELISA kit (abcam, ab210972, Cambridge, MA, USA) as per the manufacturer's instructions. Data was collected using Polar Star Omega 5.70 (BMG LabTech).

### Brain microvasculature isolation
Brain microvasculature was isolated from whole mouse brain as previously described[85]. Briefly, brains were dissected, cut into small pieces, and homogenized in MCDB131 media (Gibco, Waltham, MA, USA) with 2% Cosmic calf serum (HyClone, Logan, UT, USA) with a dounce homogenizer. Homogenates were suspended in MCDB131 media with 17% dextran (70,000 MW, Sigma-Aldrich, St. Louis, MO, USA) and centrifuged at 10,000 RPM in a swing bucket rotor. The pellet was resuspended in 1X cell lysis buffer (Cell Signaling Technology, 9803, Danvers, MA, USA) with protease inhibitors (cOmplete mini, Roche, Indianapolis, IN, USA), briefly sonicated, and centrifuged at $12,000 \times g$ at 4 °C for 15 min. Protein lysates were analyzed using either traditional Western blot or capillary electrophoresis immunoassays.

### Statistical analyses
Statistical analyses were performed using GraphPad Prism (La Jolla, CA, USA). A two-tailed, unpaired t-test was used for comparisons between means from two experimental groups. When variances were unequal, a t-test with Welch's correction was used. One-way ANOVA analyses were used as appropriate followed by Tukey's or Sidak's post-hoc tests, respectively. Normality was tested using the Brown-Forsythe analysis, when variances were unequal, a Kruskal-Wallis test was performed followed by a Dunn's multiple comparison test. $p < 0.05$ was considered significant. The Grubbs' test was used to identify statistical outliers (alpha = 0.05).

### Reporting summary
Further information on research design is available in the Nature Portfolio Reporting Summary linked to this article.

## Data availability
Source data are provided with this paper.

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

## Acknowledgements

These studies were supported by NIA 1RF1AG057964 to V.I.P. and V.G., NIA 1RF1AG068283 to V.G. and V.I.P., Merit Review Award I01 BX002211-01A2 from the US Department of Veterans Affairs Biomedical Laboratory Research and Development Service, the Robert L. Bailey and daughter Lisa K. Bailey Alzheimer's Fund in memory of Jo Nell Bailey to V.G., a William & Ella Owens Medical Research Foundation Grant, the San Antonio Medical Foundation, and the JMR Barker Foundation to V.G. These studies were also supported by an award to V.G. through the NCATS/NIH Clinical and Translational Science Award grant UL1TR002645, and by the Oklahoma Nathan Shock Center of Excellence in the Basic Biology of Aging (NIA P30AG050911), the Cellular and Molecular Geroscience Center of Biomedical Research Excellence (P20GM125528), and the Geroscience Training Program in Oklahoma (T32AG052363). We gratefully acknowledge the support of the NIH NCATS through Grant UL1TR001120, and of the Cancer Prevention and Research Institute of Texas Award RP160844 to the Center for Innovative Drug Discovery. The content is solely the responsibility of the authors and does not necessarily represent the official views of the NIH. These studies used the services of the San Antonio Nathan Shock Center of Excellence in the Biology of Aging (NIH/NIA 2 P30 AG013319-21). S.A.H. was supported by a Career Development Award (1 IK2 BX003798-01A1) from the US Department of Veterans Affairs Biomedical Laboratory Research and Development Service. C.V.S. and A.B.O. were supported by NIA Training Grant T32AG021890. C.V.S. was also supported by Alzheimer's Association AARF-17-504221. M.J.H. was supported by NCATS/NIH UL1TR001120. V.I.P. was supported by NIA/NIH R03AG052394, NIA/NIH R01AG057964, and the American Federation for Aging Research (AFAR). K.T.D. was supported by NIH/NINDS R01 NS094557. Z.U. is supported by the National Institute on Aging (R01-AG055395, R01-AG047879; R01-AG038747) and the National Institute of Neurological Disorders and Stroke (NINDS; R01-NS056218, R01-NS100782). R.K. and S.A.M. were supported by R01AG054025, RF1AG055771, and the American Heart Association. AQB was supported by NIH/NIA 1F31AG067732-01 and NIH T32GM113896 (STX-MSTP). We gratefully acknowledge Dr. Nataliya Smith for helpful contributions.

## Author contributions

V.G. conceived the studies and secured funding; S.A.H., A.Q.B., and V.G. designed the experiments; S.A.H., A.Q.B., C.V.S., A.O.D., B.F., H.M., S.F.H., M.J.H., M.G., P.A.O., S.A.M., and R.K. performed or supervised the performance of the experiments; S.A.H., C.V.S., A.Q.B., A.O.D., B.F., H.M., S.F.H., M.J.H., M.G., P.A.O., and V.G. analyzed the data; S.A.H., C.V.S., A.Q.B., M.G., P.A.O., K.T.D., Z.U., V.I.P., R.K., M.J.H., and V.G. wrote and/or edited the manuscript.

## Competing interests

R.K. is an inventor on U.S. Patent: 8,778,343 issued July 15, 2014 entitled "Antibodies that bind tau oligomers." U.S. Patent 8,778,343 is for antigen preparation, screening methods, conformational tau epitope, and corresponding antibodies (tau oligomer monoclonal antibody (TOMA). All other authors declare no competing interests.

## Additional information

[1]Department of Biochemistry and Molecular Biology, University of Oklahoma Health Sciences Center, 940 Stanton L Young Blvd, Oklahoma City, OK 73104, USA. [2]Center for Geroscience and Healthy Brain Aging, University of Oklahoma Health Sciences Center, 940 Stanton L Young Blvd, Oklahoma City, OK 73104, USA. [3]Oklahoma City Veterans Health Care System, 921 NE 13th Street, Oklahoma City, OK 73104, USA. [4]South Texas Medical Scientist Training Program, University of Texas Health San Antonio, 7703 Floyd Curl Drive, San Antonio, TX 78229, USA. [5]Barshop Institute for Longevity and Aging Studies, University of Texas Health San Antonio, 4939 Charles Katz Drive, San Antonio, TX 78229, USA. [6]Center for Therapeutic Science, University of Oklahoma Health Sciences Center, 940 Stanton L Young Blvd, Oklahoma City, OK 73104, USA. [7]Department of Molecular Medicine, University of Texas Health Science Center at San Antonio, 7703 Floyd Curl Drive, San Antonio, TX 78229, USA. [8]Departments of Neurology, Neuroscience and Cell Biology, University of Texas Medical Branch at Galveston, 301 University Blvd, Galveston, TX 77555, USA. [9]Mitchell Center for Neurodegenerative Disease, University of Texas Medical Branch at Galveston, 301 University Blvd, Galveston, TX 77555, USA. [10]Sealy Center for Vaccine Development, University of Texas Medical Branch at Galveston, 301 University Blvd, Galveston, TX 77555, USA. [11]Department of Neurosurgery, University of Oklahoma Health Sciences Center, 800 Stanton L Young Blvd, Oklahoma City, OK 73104, USA. [12]International Training Program in Geroscience, Doctoral School of Basic and Translational Medicine, Department of Public Health, Semmelweis University, H-1085 Budapest, Üllői út 26, Budapest, Hungary. [13]Hevolution Foundation, 5.08, Riyadh 13519, Saudi Arabia. [14]These authors contributed equally: Stacy A. Hussong, Andy Q. Banh. ✉e-mail: Veronica-GalvanHart@ouhsc.edu

