## [Peer Review File · Nature Communications]

Soluble pathogenic tau enters brain vascular endothelial cells and drives cellular senescence and brain microvascular dysfunction in a mouse model of tauopathyREVIEWER COMMENTS

Reviewer #1 (Remarks to the Author):

In this manuscript entitled, "Soluble pathogenic tau enters brain vascular endothelial cells and drives cellular senescence and brain microvascular dysfunction in tauopathy" authors have investigated the role of microvasculature in tauopathy.

There are several technical aspects that authors must address before this manuscript can be accepted for publication:

Major concern:

- 1. Author have assumed that lectin staining can and will stain the vasculature/endothelial cells in the brain. However, if the animals are not perfused well, one will not see all the vasculature in the brain.**
- 2. Moreover, lectin tends to stain microglia or macrophage if the brain is heavily inundated with plaques.**
- 3. The photomicrographs should be of high quality to decipher the role of microvasculature in Tauopathy. Unfortunately, none of immunofluorescence slides/images are of high quality.**
- 4. Authors can use PECAM/CD31 (BD Pharmingen) which stains specifically endothelial cells in the mouse brain. Alternatively, authors could use Collagen IV (Abcam) to visualize microvessel in the brain.**
- 5. There is cross bleeding between the filters (figure 1B). Please provide light microscopic images of Tau5 and Tau 22 in both wild type and Tg animals.**
- 6. It is very difficult to comprehend which areas of the brain has been studied.**
- 7. Did authors want to investigate cerebrovascular/endothelial dysfunction in tauopathy model? If so, authors have to more specific whether any specific region is more vulnerable to study.**
- 8. Authors have refereed reference 18 in which they did double immunolabeling in the human brain. However, even in that manuscript it was not well described whether they looked at meningeal vessels, capillaries or in the cerebral arteries or capillaries. Based on the location, the function of the vessels may vary, so is their modus operandi.**
- 9. More importantly, when they study in the animal model, they need to describe when tauopathy starts, which areas of the brain they see them first and how it progresses over time.**
- 10. It is very difficult to fathom from the materials and methods what are the age group of the animals. Based on the age of the animals, microvascular permeability, dysfunction will be altered.**
- 11. Authors wrote, "Animals were euthanized by cervical dislocation". This is not the best method to detect to do any IHC be it for light microscopy or fluorescence microscopy, thus, authors are encouraged to do additional study dedicating n=8 animals just for IHC.**
- 12. Above method is crude and have lots of blood borne artifacts that will interfere the results all along.**
- 13. Vascular reactivity experiment was done meticulously.**
- 14. Based on the in vitro experimens they suggest that similar to pathogenic tau entry to neurons, internalization of soluble tau aggregates promotes phosphorylation of endogenous tau in human brain vascular endothelial cells.**
- 15. Authors wrote, "Removal of soluble tau aggregates with immunotherapy reduces levels of tau in brain microvasculature and ameliorates brain vascular endothelial cell dysfunction in P301S(PS19) mice". In this regard, authors are encouraged to write in detail about immunotherapy of mice.**
- 16. Authors wrote, "Brain vascular density was unchanged in 12 month-old P301S(PS19) mice as compared to aged matched controls (p=0.43, Figure 6G-H), ruling out a potential impact of decreased vascular density on functional outcomes (Figure 6A-B). Taken together, these data indicate that soluble pathogenic tau aggregates contribute significantly to brain microvascular dysfunction in a model of tauopathy". Does it mean**

that there was no change in the cortical vascular density? No endothelial dysfunction? It would not harm to do a simple Prussian Blue stain to detect hemosiderin if there was any leakage or ischemia due to Tauopathy.

17. It would be nice if authors take a bit more time tell about the impact of Tau on other neurovascular units in the brain and describe about the role of pericytes, basement membranes and gliovascular coupling as well.

Reviewer #2 (Remarks to the Author):

The work by Hussong et al. is a straightforward study to examine the role of soluble pathogenic forms of tau in facilitating brain microvascular deficits and brain microvascular endothelial cell dysfunction and senescence. Soluble tau aggregates were accumulated in brain microvasculature and were internalized by brain microvascular endothelial cells, promoting phosphorylation of endogenous tau. Soluble tau aggregates in endothelial cells decreased eNOS activity, triggered endothelial senescence. Most importantly, removal of soluble tau aggregates by tau oligomer-specific monoclonal antibody restored endothelial function, thus suggesting therapeutic potential in treatment of Alzheimer's disease and other tauopathies. This work adds interesting and potentially important new insight into the pathological role of tau. However, it still exists some minor questions.

1. The present observations were mainly in P301S(PS19) mice model. How about other AD animal models? Do soluble tau aggregates accumulate in brain microvasculature in other AD models?

2. Do soluble tau aggregates enter brain microvascular smooth muscle cells? Do they affect the contractile function of smooth muscle? Stimulators such as sodium nitroprusside should be used to elicit endothelium-independent relaxations in order to study the possible effect of soluble tau aggregates on brain microvascular smooth muscle cells.

3. In Figure 1B, the fluorescence signal of Tau5 seems to be comparable between wildtype mice and P301S(PS19) mice. However, it is quite different in Figure 1C measured by capillary electrophoresis immunoassays. Could the authors explain the discrepancy?

4. Markers for senescent cells such as expression of senescence-associated- β -galactosidase should be detected.

5. Although eNOS activation and inhibition were studied by phosphorylation at S1177 and T495, NO production should be measured by DAF-FM staining and Griess assay.

6. The authors indicated that microtubule destabilization induced by soluble tau aggregates might affect the transport of eNOS to membrane compartments. However, the evidence was not solid. It's better to provide visible evidence to show the translocation of eNOS.

7. Tau oligomer-specific monoclonal antibody demonstrated protective effects against endothelial dysfunction and microvascular deficits. Does it alleviate pathological changes and symptoms of AD like cognitive impairment?

Reviewer #3 (Remarks to the Author):

Soluble pathogenic tau enters brain vascular endothelial cells and drives cellular senescence and brain microvascular dysfunction in tauopathy

This paper builds on the author's previous demonstration that tau oligomers accumulate in microvascular endothelial cells in AD and other tauopathies. The current study shows similar findings in P201S tau mice and demonstrates that tau oligomers use a heparan sulfate proteoglycan dependent pathway to enter brain microvascular cells, which is followed by marked endothelial cell dysfunction including acquisition of the senescence

associated secretory phenotype (SASP). The study provides novel information that is of broad interest.

Please provide a higher power confocal view of Fig. 1b so that the location of the vascular accumulation of tau in P301S mice can be better ascertained. Electron micrographs would be particularly useful. Also, please provide a low power view of the entire brain section so that regional accumulation of vascular tau can be ascertained.

Does endothelial dysfunction require endogenous endothelial tau expression, as could be demonstrated through tau siRNA experiments? The data presented do not define a causal pathway, as both internalized tau as well as endogenous tau could contribute to reduced eNOS activation.

SASP markers are different in vitro vs in vivo, with MCP-1 being the most upregulated in vivo with no change in IL-6, whereas the opposite pattern is observed In vitro where IL-6 is upregulated ~25 fold. The authors suggest that a lower incidence of in vivo uptake and the influence of other cell types in the in vivo neurovascular unit may explain the discrepancy. This is unsatisfactory as a lower incidence should reduce the levels of SASP markers but not alter the pattern of activation, and the influence of other cells types has not been experimentally addressed, such as with transwell assays that include pericytes and astrocytes.

The immunotherapy experiments do not provide mechanistic insights. For example, it is possible that endothelial function is restored indirectly via improved neuronal function, neurovascular coupling, and vascular reactivity. Also, TOMA treatment would also reduce tau found in the blood, and the peripheral vs CNS contributions of tau on endothelial cell dysfunction cannot be distinguished by the present experiments. These limitations should be acknowledged.

Methods are appropriate and sufficiently detailed. With respect to statistical methods, please provide additional information for meeting ANOVA criteria (i.e. normality) when this test is used.

Cheryl Wellington

RESPONSE TO THE REVIEWERS

(1) **Reviewer #1** (Remarks to the Author):

“In this manuscript entitled, “Soluble pathogenic tau enters brain vascular endothelial cells and drives cellular senescence and brain microvascular dysfunction in tauopathy” authors have investigated the role of microvasculature in tauopathy.

There are several technical aspects that authors must address before this manuscript can be accepted for publication:

Major concern:

R1 Comment #1. *“.. Authors have assumed that lectin staining can and will stain the vasculature/endothelial cells in the brain. However, if the animals are not perfused well, one will not see all the vasculature in the brain.”*

We thank the reviewer for this important comment. We agree that our studies using lectin may not reveal the entirety of brain microvasculature nor the full extent of vascular tau pathology in P301S(PS19) mice. Our goal for the studies reported, however, was to determine whether vascular pathogenic tau accumulation observed in human AD and in other tauopathies (our studies, Castillo-Carranza et al 2017 PMC5440106) was recapitulated in the P301S(PS19) mouse model. Studies that will precisely map and define the full extent of vascular tauopathy in P301S(PS19) mice and in a second model of human tauopathy (Htau mice) are currently ongoing in our lab.

R1 Comment #2. *“.. Moreover, lectin tends to stain microglia or macrophage if the brain is heavily inundated with plaques.”*

We thank the reviewer for this comment. We should clarify that our studies used P301S(PS19) mice that model tauopathy alone and are devoid of amyloid plaques. However, to address the reviewer's concern that lectin may stain microglia or infiltrating macrophages we performed double-labeling immunohistochemistry studies that used fluorescently labeled lectin together with an anti-CD31 antibody (PECAM/CD31, BD Pharmingen) as suggested by the reviewer.

Our studies show that fluorescently-labeled lectin stained CD31-immunoreactive structures exclusively (i.e. no lectin-positive signals were found that were not co-localized with CD31 immunoreactivity, **Supplementary Figure 4A** in the revised manuscript) in brains of P301S(PS19) mice, indicating that lectin does not highlight microglia or infiltrated macrophages in brains of this mouse model at the ages used in our studies. The text of the legend to **Supplementary Fig. 4A** in the revised manuscript has been modified to include this information as follows:

.. Representative images of cortical brain sections show that lectin stains endothelial cells of the brain microvasculature only, as shown by its exclusive co-localization with CD31 immunoreactivity.”

R1 Comment #3. *“.. The photomicrographs should be of high quality to decipher the role of microvasculature in Tauopathy. Unfortunately, none of immunofluorescence slides/images are of high quality.”*

We agree with the reviewer. As per the journal's submission guidelines, we included lower-resolution versions of our figures, that had to be embedded in the text, in the initial submission of our manuscript. Unfortunately at this stage in the submission process, we are still unable to upload high resolution figures. We would be delighted to provide high resolution figures as needed. If the reviewer would like to obtain the high resolution figures, she/he should please contact the handling editor.

R1 Comment #4. *“.. Authors can use PECAM/CD31 (BD Pharmingen) which stains specifically endothelial cells in the mouse brain. Alternatively, authors could use Collagen IV (Abcam) to visualize microvessel in the brain.”*

We thank the reviewer for this important suggestion. As mentioned in our response to Comment #2, as per the reviewer's suggestion we performed immunohistochemistry studies in brain sections using the anti-CD31

PECAM/CD31 antibody produced by BD Pharmingen in combination with fluorescently labeled lectin. These studies revealed that fluorescent lectin exclusively stains CD31 immunoreactive-material in brain tissues from P301S(PS19) mice (**Supplementary Figure 4A** in the revised manuscript). We did not detect any lectin signals that did not co-localize with CD31 immunoreactivity in brains of P301S(PS19) mice. These data indicate that fluorescently labeled lectin can serve as a useful marker for brain microvasculature in P301S(PS19) mice modeling tauopathy.

R1 Comment #5. “.. There is cross bleeding between the filters (figure 1B). Please provide light microscopic images of Tau5 and Tau 22 in both wild type and Tg animals.”

We thank the reviewer for bringing up this important point, relevant to data in **Figure 1B** (now **Figure 1A**). To rule out spectral bleed-through in our confocal imaging studies we routinely collect each channel separately by sequential scanning of the specimen with individual lasers and detecting fluorescence in each channel to coincide with laser illumination (please see point 2a below).

Bleed-through could have occurred (1) from the Tau5 channel to the T22 channel and (2) from the T22 channel to the Tau5 channel. To further address the reviewer’s concern, each scenario is considered below:

(1) Bleed-through from T5 to T22. We acquired images from all experimental groups using identical settings in a confocal microscope. Therefore, if present, bleed-through from T5 to T22 would be detected in all images, including those obtained from WT animals. T5 immunofluorescent signals arising from endogenous mouse tau expression were present in brain sections from WT mice (**Figure 1A**), but no signals were detected in the T22 channel (**Figure 1A**). Thus, bleed-through from T5 to T22 can be ruled out.

(2) Bleed-through from T22 to Tau5: (2a) Bleed-through generally occurs between neighboring channels, most commonly from lower wavelength channels to higher wavelength channels (e.g. green → red → far-red). Although we routinely collect each channel separately, to further ensure the absence of bleed-through between T22 and Tau5 signals we also routinely use fluorophores whose spectra are non-overlapping. Because there is no physical overlap between AlexaFluor488 and AlexaFluor647 spectra, T22 was imaged in the green channel (via AlexaFluor488-labeled specific secondary antibodies) and Tau5 was imaged in the far red channel (via AlexaFluor647-labeled specific secondary antibodies). (2b) Further, in the unlikely case that the T22 (AlexaFluor288, green channel) signal had bled-through to the Tau5 channel (AlexaFluor647, far red), the T22 bleed-through signal would be found in both the red (DyLight 594-lectin) and the far-red (Tau5) channel. Since T22 bleed-through is not observed in the red channel (red channel image in **Figure 1A**), overlap in the images due to bleed-through between T22 and Tau 5 can be completely ruled out.

The partial colocalization of signals from T22 and Tau5 antibodies is expected, because T22 recognizes a subset of tau molecules (soluble tau aggregates) while Tau5 recognizes all forms of tau.

R1 Comment #6. “.. It is very difficult to comprehend which areas of the brain has been studied.”

We thank the reviewer for this important comment. We have modified the text in the revised manuscript such that the brain region that was used in each of our studies is indicated in the **Methods** section, the **Results** section or in the **Figure legends** as follows:

Methods (page 22 line 561-564)

“.. The bone over the left somatosensory barrel cortex (0.5-1.5 mm posterior and 3-4.5 mm lateral from bregma) was removed, a silicone barrier was made around the craniotomy, aCSF was applied, and a laser Doppler probe (Transonic, Ithaca NY, USA) was positioned over the barrel cortex.”

Methods (page 23, line 585)

“.. Brain microvasculature was isolated from whole mouse brain...”

Methods (page 18, line 452-453)

“.. To quantify tau oligomers (soluble tau aggregates) by traditional Western blot, brain tissue lysates were generated from prefrontal cortex...”

Results (page 10, line 228-229)

“ .. Consistent with our observations in human AD brain, we found substantial accumulation of soluble oligomeric tau aggregates in cortical brain microvasculature of 8-month-old P301S(PS19) mice (**Figure 1A**)...”

Results (page 11, line 270-271)

“.. As expected, TOMA treatment significantly decreased the amount of soluble tau aggregates in the cortex of 12-month-old P301S(PS19) mice as compared to untreated controls ($p < 0.05$, **Figure 6C-D**).”

Results (page 11, line 271-274)

“ .. In agreement with these observations, we found that cortical microvascular tau deposits in this cohort were increased in P301S(PS19) untreated animals ($p = 0.026$, **Figure 6E-F**), but not in TOMA-treated P301S(PS19) mice, as compared to WT controls (**Figure 6E-F**).”

Figure 1 legend (page 30, line 836-838)

“ .. Representative images of cortical brain sections from 8-month-old P301S(PS19) and WT controls showing soluble tau aggregates (T22, green) and total tau (Tau5, cyan) immunoreactivity in lectin-stained microvasculature (red).”

Figure 5 legend (page 37, line 904-905)

“ .. Representative immunofluorescent images of cortical brain sections from WT control and P301S(PS19) mice stained with DAPI (blue) and lectin (red) and immunostained for acetyl- α -tubulin (green)

Figure 6 legend (page 38, line 927-928)

“.. Treatment with TOMA ameliorates profound deficits in endothelium-dependent vascular responses in somatosensory cortex of P301S(PS19) mice.”

Figure 6 legend (page 38, line 936-937)

“Representative immunoblots of cortical lysates of P301S(PS19) mice treated with isotype-matched IgG or TOMA”

Figure 6 legend (page 38, line 940-942)

“Representative images (100X) of tau immunoreactivity (green) associated with lectin-reactive cortical brain vasculature (red), counterstained with DAPI (blue) staining.”

Supplementary Figure 4 legend (page 42, line 981-989)

“**A**, Representative images (63x) of cortical brain sections show that lectin precisely stains endothelial cells of the brain microvasculature as shown by exclusive colocalization with CD31. **B**, Representative low-magnification images (10x) of cortical brain sections showing total tau (Tau5, green) immunoreactivity in P301S(PS19) and WT brain microvasculature (lectin, red) at 8 months of age. **C**, Representative high-magnification images (63x+2.5x digital zoom) of cortical brain sections showing total tau total tau (Tau5, green) immunoreactivity P301S(PS19) and WT brain endothelial cells (CD31, red) at 8 months of age. **D**, Representative images (63x) of brain microvasculature (lectin, red) and fibrinogen (green) cortical brain sections of P301S(PS19) and WT at 8 months of age. ”

Supplementary Figure 5 legend (page 43, line 994-998)

“**Supplemental Figure 5**. Treatment with tau oligomer-specific monoclonal antibody (TOMA) does not reduce cortical brain fibrillar tau or total plasma tau in 12 month-old P301S(PS19) mice. **A**, Representative images (63x) of cortical brain sections showing fibrillar tau immunoreactivity (phospho-tau, AT8, green) in 12 month-old WT, P301S(PS19), and P301S(PS19) mice treated with tau oligomer-specific monoclonal antibody (TOMA).”

R1 Comment #7. “.. *Did authors want to investigate cerebrovascular/endothelial dysfunction in tauopathy model? If so, authors have to more specific whether any specific region is more vulnerable to study.*”

We thank the reviewer for this comment. *In vivo* approaches to measure brain cerebrovascular/endothelial function in live animals have been established and validated for laser Doppler flowmetry, and for *in vivo* 2-photon imaging of the somatosensory cortex. Comparison of functional vulnerability among different areas of the brain would be limited to the uppermost cortical layers, thus would not be informative.

Other *in vivo* studies of vascular reactivity that could also potentially inform the reviewer's question would require the use ASL magnetic resonance imaging approaches. The interpretation of these MRI-based studies, however, would be limited by the need to deliver potent vasodilatory stimuli systemically (as opposed to their local, topical application in laser Doppler flowmetry and in 2-photon studies, where all influences from systemic responses are avoided).

We undertook studies to clarify the reviewer's questions through biochemical analysis of NO release in tissue homogenates from specific brain regions of P301S(PS19) and WT mice after systemic acetylcholine stimulation (please see response to **Reviewer 2 Comment # 5**). Unfortunately, studies in control WT animals of the P301S(PS19) line treated acutely with a wide range of doses of ACh did not reveal changes, suggesting that the activation of eNOS and NO release in WT and P301S mice is transient and that the system likely returns to baseline by the time the animals are sacrificed, their brains are dissected, and homogenates are generated. Thus, unfortunately we were not able to compare vulnerability of different brain regions to tau-induced cerebrovascular/endothelial dysfunction. ASL MRI-based studies in P301S(PS19) and in a second mouse model of tauopathy, hTau mice, are planned and will be performed in the future.

R1 Comment #8. *".. Authors have referred reference 18 in which they did double immunolabeling in the human brain. However, even in that manuscript it was not well described whether they looked at meningeal vessels, capillaries or in the cerebral arteries or capillaries. Based on the location, the function of the vessels may vary, so is their modus operandi."*

We thank the reviewer for this important question. The study by Castillo-Carranza et al (Castillo-Carranza et al 2017 PMC5440106) measured total tau (with Tau5, a pan-reactive anti-tau antibody) and oligomeric tau (with T22, an antibody specific for soluble tau oligomers) in cortical capillaries and arterioles. We apologize about this omission. We have added this information in the **Results** section of the revised manuscript as follows:

Results (page 5, line 106-108)

".. Pathogenic soluble tau aggregates (tau oligomers) accumulate in cortical capillaries and arterioles of AD and PSP¹⁸ (secondary and primary tauopathies respectively) in association with brain microvascular endothelial cells¹⁸."

R1 Comment #9. *".. More importantly, when they study in the animal model, they need to describe when tauopathy starts, which areas of the brain they see them first and how it progresses over time."*

We thank the reviewer for pointing out this omission. To correct it, we have updated the Methods section (page 21, line 527-532) to include the following information:

Methods (page 21, line 526-531):

".. At 3 months of age total tau (but not hyperphosphorylated tau) increases in P301S(PS19) mice, together with synaptic loss and microgliosis in hippocampus, amygdala, cortex, brain stem, and spinal cord. Inflammatory cytokines increase in hippocampus at 4 months of age. By 6 months of age, total tau and hyperphosphorylated tau increase in the same brain regions accompanied by neurofibrillary tangles and increased astrogliosis. Neuron loss accompanied by hippocampal and cerebral cortical atrophy begins at 9 months of age (Yoshiyama et al 2007 PMID: 17270732)."

R1 Comment #10. *".. It is very difficult to fathom from the materials and methods what are the age group of the animals. Based on the age of the animals, microvascular permeability, dysfunction will be altered."*

We thank the reviewer for bringing up this important point. We have added the age of the animals that were used for all the studies reported in the corresponding **Figure legends** of the revised manuscript. Overall animals used in our studies were 8 or 12 months of age. A group of 4 month-old mice were used in the studies reported in **Figure 4A**.

R1 Comment #11. *".. Authors wrote, "Animals were euthanized by cervical dislocation". This is not the best method to detect to do any IHC be it for light microscopy or fluorescence microscopy, thus, authors are encouraged to do additional study dedicating n=8 animals just for IHC. Above method is crude and have lots of blood borne artifacts that will interfere the results all along."*

We thank the reviewer for this comment. We have perfected our approach to cervical dislocation of animals at a deep plane of anesthesia at the end of brain surgeries during laser Doppler flowmetry such that no artifacts

are generated by the procedure. Cardiopulmonary arrest occurs immediately after the cervical spinal cord has been severed at multiple sites by the procedure. Cardiopulmonary arrest is verified before the brain is dissected out and flash frozen, all within 60-80 seconds from the time of cardiopulmonary arrest. We have verified that our approach effectively circumvents vascular or other tissue artifacts in all our experiments using immunohistochemistry for a variety of vascular and tissue integrity markers. These studies did not detect any vascular or parenchymal abnormalities as a result of cervical dislocation in WT groups or in several different transgenic animals modeling Alzheimer's tauopathy or amyloidopathy (Lin et al 2013 PMC3764385, Lin et al 2015 PMC5167110, Van Skike et al 2018 PMC5966773, Jahrling et al 2018 PMC5757441, Van Skike et al 2020 PMC6974719, Van Skike et al 2021 PMC8143195). This is also the case for the studies reported in our manuscript (**Fig 1A**, **Supplementary Fig 4**, **Fig 5A**, **Fig 6E** and **G**, and **Supplementary Fig 5A-B**).

A paragraph addressing this point has been added to the Methods section (page 21, line 533-539) in the revised manuscript, as follows:

Methods section (page 21, line 533-539)

".. Cardiopulmonary arrest is verified before the brain is dissected out, within 60-80 seconds after cervical dislocation. We have confirmed that our approach effectively circumvents vascular or other tissue artifacts in all our experiments using immunohistochemistry for a variety of vascular markers. These studies have not detected any vascular or parenchymal abnormalities in wild-type non-transgenic or transgenic groups as a result of cervical dislocation (Lin et al 2013 PMC3764385, Lin et al 2015 PMC5167110, Van Skike et al 2018 PMC5966773, Van Skike et al 2020 PMC6974719, Van Skike et al 2021 PMC8143195) including the studies reported in the present manuscript."

R1 Comment #12. "*.. Vascular reactivity experiment was done meticulously.*"

We thank the reviewer for her/his comment. We carefully control and monitor our experiments to minimize variance and avoid common pitfalls.

R1 Comment #13. "*.. Based on the in vitro experiments they suggest that similar to pathogenic tau entry to neurons, internalization of soluble tau aggregates promotes phosphorylation of endogenous tau in human brain vascular endothelial cells.*"

We agree with the reviewer's comment. Our data shows that tau is internalized by endothelial cells (**Figure 1B**) in a manner similar to the internalization of pathogenic tau by neurons.

R1 Comment #14. "*.. Authors wrote, "Removal of soluble tau aggregates with immunotherapy reduces levels of tau in brain microvasculature and ameliorates brain vascular endothelial cell dysfunction in P301S(PS19) mice". In this regard, authors are encouraged to write in detail about immunotherapy of mice.*"

We thank the reviewer for this suggestion. We have included a detailed description of tau oligomer monoclonal antibody (TOMA) immunotherapy in P301S(PS19) mice in the Methods section (page 22-23, line 576-581) and the Results section (page 10, line 242-243) and cite appropriate references to prior work validating this approach (references 24, 52, 53, 67) in the revised manuscript, as follows:

Methods section (page 22-23, line 576-581)

"WT and P301S(PS19) mice were treated with 60 µg of TOMA⁵⁵ or same-isotype IgG (Rockland Immunochemicals, Limerick, PA, USA) via tail vein injection, every 2 weeks starting after disease onset at 5.5 months of age and continuing for 8 weeks until 7.5 months of age. TOMA was developed and validated in the Kaye laboratory⁵⁵, where dose and regimen for TOMA immunotherapy was established^{24,54,55,79}, with no discernible side effects in P301L and Htau mice^{24,55}."

Results section (page 11, line 259-260)

".. Tau oligomer-specific monoclonal antibody (TOMA, 24,52,53) recognizes soluble tau aggregates but does not recognize soluble, functional forms of tau or tau fibrils."

R1 Comment #15. “.. Authors wrote, “Brain vascular density was unchanged in 12 month-old P301S(PS19) mice as compared to aged matched controls ($p=0.43$, Figure 6G-H), ruling out a potential impact of decreased vascular density on functional outcomes (Figure 6A-B). Taken together, these data indicate that soluble pathogenic tau aggregates contribute significantly to brain microvascular dysfunction in a model of tauopathy”. Does it mean that there was no change in the cortical vascular density? No endothelial dysfunction? It would not harm to do a simple Prussian Blue stain to detect hemosiderin if there was any leakage or ischemia due to Tauopathy.”

We thank the reviewer for this comment. We did find profound endothelial dysfunction, manifested as absent acetylcholine-stimulated vascular reactivity in somatosensory cortex of 12-month-old P301S(PS19) mice (Figure 6A,B). The observed deficits in vascular reactivity, however, were not due to an overall loss of density in the cortical vascular network, since we found no detectable changes in cortical vascular density in 12 month-old P301S(PS19) mice (Figure 6G,H). To clarify this point we have changed the text of the Results section (page 12, line 291-293), as follows:

Results (page 12, line 291-293)

“.. Brain vascular density was unchanged in 12 month-old P301S(PS19) mice as compared to aged matched controls ($p=0.43$, Figure 6G-H), ruling out a potential impact of decreased vascular density on functional outcomes (Figure 6A-B). “

While endothelial dysfunction was prominent in P301S(PS19) mice, to answer the reviewer’s question we used immunohistochemistry to measure fibrinogen present in vasculature or in brain parenchyma, the latter a marker of blood-brain barrier leakage, in P301S(PS19) mice. We found no significant difference in blood-brain barrier permeability (as amount of fibrinogen in brain parenchyma) in P301S(PS19) animals when compared to age-matched WT littermate controls. These data are shown in **Supplementary Figure 4D,E** and are described in the Results section (page 10, line 231-233) of the revised manuscript as follows:

Results (page 10, line 231-233)

“.. The accumulation of pathogenic soluble tau in brain microvasculature of P301S(PS19) mice, however, was not associated with leakage arising from blood-brain barrier dysfunction (Supplementary Figure 4D-E).”

R1 Comment #16. “... It would be nice if authors take a bit more time tell about the impact of Tau on other neurovascular units in the brain and describe about the role of pericytes, basement membranes and gliovascular coupling as well.”

We appreciate the reviewer’s suggestion. To address the reviewer’s comment, we performed additional studies that revealed that, in addition to inducing profound microvascular endothelial cell dysfunction, pathogenic tau also negatively impacts smooth muscle cells, another component of the neurovascular unit (**Supplementary Figure 3** in the revised manuscript). We have modified the text of the Results section (page 9, line 218-225) and the Discussion section (page 13, line 31-315) in the revised manuscript respectively as follows:

Results section (page 9, line 218-225)

“Responses to SNP in P301S(PS19) mice were significantly reduced as compared to those of WT littermates (**Supplementary Figure 3**), indicating that tauopathy is associated with reduced smooth muscle function in P301S(PS19) mice. While brain vascular endothelium responses in P301S(PS19) mice were completely ablated (Figure 4A), however, the response of smooth muscle to SNP was reduced but not absent, indicating that tau-induced smooth muscle injury is of lesser magnitude than tau-induced endothelial injury in P301S(PS19) mouse brain microvasculature. Tau-induced smooth muscle injury may thus contribute to but does not explain the absence of vasodilatory responses in brains of P301S(PS19) mice (Figure 4A).”

Discussion section (page 14, line 335-338).

“Moreover, and consistent with our previous studies in AD and PSP brain that showed accumulation of pathogenic soluble tau in brain microvascular smooth muscle cells¹⁸, our data show that pathogenic soluble tau aggregates also decrease smooth muscle function in P301S(PS19) mice. ”

In addition, we have modified the Discussion section (page 14, line 338-346) in the revised manuscript to describe the potential impact of pathogenic tau on other components of the neurovascular unit, including pericytes and astroglia as follows:

Discussion section (page 14, line 338-346)

“ While our data indicate that the impact of pathogenic tau on smooth muscle is of lesser magnitude than its impact on endothelial cell function, these observations suggest that the etiology of pathogenic tau microvascular dysfunction is multifaceted and likely involves multiple brain microvascular cell types that compose the neurovascular unit (including possibly astrocytes and pericytes) and likely not limited to endothelium and smooth muscle cells. Whether detriment to smooth muscle cell function by soluble aggregated tau in P301S(PS19) mice is dependent or independent of its actions on microvascular endothelial cells cannot be ascertained from our data. Future studies will clarify this distinction, and also define the contribution of pathogenic tau-induced damage in astrocytes and pericytes to brain microvascular dysfunction in AD and other tauopathies.”

(2) Reviewer #2 (Remarks to the Author):

“The work by Hussong et al. is a straightforward study to examine the role of soluble pathogenic forms of tau in facilitating brain microvascular deficits and brain microvascular endothelial cell dysfunction and senescence. Soluble tau aggregates were accumulated in brain microvasculature and were internalized by brain microvascular endothelial cells, promoting phosphorylation of endogenous tau. Soluble tau aggregates in endothelial cells decreased eNOS activity, triggered endothelial senescence. Most importantly, removal of soluble tau aggregates by tau oligomer-specific monoclonal antibody restored endothelial function, thus suggesting therapeutic potential in treatment of Alzheimer’s disease and other tauopathies. This work adds interesting and potentially important new insight into the pathological role of tau. However, it still exists some minor questions.

R2 Comment #1. *“.. The present observations were mainly in P301S(PS19) mice model. How about other AD animal models? Do soluble tau aggregates accumulate in brain microvasculature in other AD models?”*

We thank the reviewer for this comment. Our previous studies showed that pathogenic soluble tau accumulates in brain vasculature of Alzheimer’s disease (AD) patients and in other tauopathies (Castillo-Carranza et al 2017 PMC5440106). In this publication, we also reported the accumulation of soluble pathogenic tau in brain vasculature of the Tg2576 model of AD amyloidopathy. In addition to the studies included in the present manuscript using P301S(PS19) mice, our unpublished data show that soluble aggregated tau also accumulates in brain microvasculature of aged marmosets, a non-human primate species (Figure shown below, panels **C** and **D**) and that Thr213-phosphorylated tau (Thr213 is a pathogenic phosphorylation event that precedes tau misfolding and aggregation) accumulates in brain microvasculature of hTau mice, a model of AD tauopathy (Figure shown below, panels **A** and **B**). Taken together, these data suggest that brain microvascular pathogenic tau accumulation is found in 2 primate species and in several mouse models of AD, including models of amyloidopathy (Tg2576) and tauopathy (P301S and hTau mice).

Brain vascular tau accumulation in hTau mice modeling AD tauopathy and in aged marmoset monkeys (*Callithrix jacchus*). A-B, Accumulation of Thr231-phospho human tau in brain microvasculature of hTau mice. A, Representative images of 9 month-old WT and hTau mice cortex stained with fluorescently-labeled tomato lectin (green) to highlight microvasculature and with AT180 (Thr231-Tau, red) antibody; B, Quantitative analyses of data in A. C-D, Accumulation of soluble tau aggregates in brain microvasculature of aged (12 years-old) marmosets; C, Representative images of prefrontal cortex (PFC) of young (4 years-old) and aged (12 years-old) marmosets stained with fluorescently-labeled (red) tomato lectin to highlight microvasculature and with T22 antibodies specific for aggregated soluble tau (green). \$\$\$, $P < 0.001$; *, $P < 0.05$, unpaired Student's t test. Data are representative images and means \pm SEM.

R2 Comment #2. “Do soluble tau aggregates enter brain microvascular smooth muscle cells? Do they affect the contractile function of smooth muscle? Stimulators such as sodium nitroprusside should be used to elicit endothelium-independent relaxations in order to study the possible effect of soluble tau aggregates on brain microvascular smooth muscle cells.”

We thank the reviewer for this important question. Although we did not measure uptake of tau in brain vascular smooth muscle cells, we previously showed that soluble tau aggregates accumulate in vascular smooth muscle cells of Alzheimer's disease brains (Castillo-Carranza et al., 2017 PMC5440106), suggesting that soluble tau aggregates may be internalized by brain microvascular smooth muscle cells.

To further explore this observation and answer the reviewer's question, we measured smooth muscle cell function in 8-month-old P301S(PS19) mice using sodium nitroprusside (SNP), an NO donor. Our studies showed that in addition to endothelial dysfunction, smooth muscle function is impaired in 8 month-old P301S(PS19) mice overexpressing mutant human tau (**Supplementary Figure 3** in the revised manuscript).

Of note, while we found that the response of brain vascular endothelium in P301S(PS19) mice was completely ablated (as shown by vasoconstriction through direct action of ACh on smooth muscle in the absence of endothelium contributions, **Figure 4A**), the response of smooth muscle of P301S(PS19) mice to SNP was decreased but not ablated, suggesting that pathogenic tau-induced smooth muscle cell dysfunction is of lesser magnitude than pathogenic tau-induced injury to brain endothelium in P301S(PS19) mice.

Taken together with our studies of vascular reactivity to acetylcholine (**Figure 4A** and **Figure 6A-B** in the revised manuscript) our data suggest that both brain vascular endothelium and smooth muscle are impaired, albeit to different degrees, by pathogenic human tau in P301S(PS19) mice modeling tauopathy. The impairment in smooth muscle function likely contributes to impaired vasodilation to acetylcholine. However, the

paradoxical vasoconstriction to acetylcholine stimulation in P301S(PS19) mice, indicative of completely absent endothelial response, cannot be explained by the partial impairment of smooth muscle alone (i.e. if endothelium were not impaired, the decrease in vascular reactivity to ACh in P301S(PS19) mice would be equal to the decrease in smooth muscle function).

The text of the Results section (page 9, line 216-225) and the text of the Discussion section (page 13, line 311-315) have been modified in the revised manuscript to include this information and a paragraph discussing these observations respectively, as follows:

Results (page 9, line 217-225)

“.. Thus, to define a potential impact of pathogenic soluble tau aggregates on vascular smooth muscle cells we assessed the response of cortical microvasculature to sodium nitroprusside (SNP53), a nitric oxide donor. Responses to SNP in P301S(PS19) mice were significantly reduced as compared to those of WT littermates (**Supplementary Figure 3**), indicating that tauopathy is associated with reduced smooth muscle function in P301S(PS19) mice. While brain vascular endothelium responses in P301S(PS19) mice were completely ablated (**Figure 4A**), however, the response of smooth muscle to SNP was reduced but not absent, indicating that tau-induced smooth muscle dysfunction is of lesser magnitude than tau-induced endothelial injury in P301S(PS19) mouse brain. Tau-induced smooth muscle injury may thus contribute to but does not explain the absence of vasodilatory responses in brains of P301S(PS19) mice (**Figure 4A**).”

Discussion (Page 13-14 , line 332-338)

“Moreover, and consistent with our previous studies in AD and PSP brain that showed accumulation of pathogenic soluble tau in brain microvascular smooth muscle cells(18), our data show that pathogenic soluble tau aggregates also decrease smooth muscle function in P301S(PS19) mice.”

R2 Comment #3. *“In Figure 1B, the fluorescence signal of Tau5 seems to be comparable between wildtype mice and P301S(PS19) mice. However, it is quite different in Figure 1C measured by capillary electrophoresis immunoassays. Could the authors explain the discrepancy?”*

We thank the reviewer for this question. Tau signal in immunohistochemistry studies of Figure 1B (**Figure 1A** in the revised manuscript) is representative of tau present in microvasculature of whisker barrel cortex in WT and P301S(PS19) mice. Wes immunoassay data in Figure 1C (**Figure 4B** in the revised manuscript), on the other hand, are representative of overall brain microvascular tau since the material probed was brain microvasculature isolated from whole brain. While both mouse tau and transgenic human tau are expressed in every neuron in P301S(PS19) mice, it has been shown transgenic human tau is expressed at different levels in different brain areas. Moreover, it is possible that differences in epitope availability in postfixed brain sections in immunohistochemistry studies of **Figure 1A** (in the revised manuscript) as compared to denatured lysates prepared from whole brain microvasculature in Wes immunoassay studies of **Figure 4B** (in the revised manuscript) may contribute to the apparent discrepancy in tau levels detected in WT and P301S(PS19) mice using these two different approaches.

R2 Comment #4. *“.. Markers for senescent cells such as expression of senescence-associated- β -galactosidase should be detected.”*

We thank the reviewer for this comment. It has been shown (Gorgoulis et al 2019 PMID: 31675495) that β -galactosidase (β -gal) staining is not a required component of senescence and can be absent in senescent cells. β -galactosidase has a very strict pH requirement and highly variable times for development of the colorimetric reaction, that need to be established for each type of cell or tissue under study. These factors lead to frequent false positives. For these reasons, the International Cell Senescence Association (Gorgoulis et al 2019 PMID: 31675495) recommended that β -gal staining be used only for exploratory purposes. Confirmation of cellular senescence requires (a) evidence of upregulation of p16 and p21 (upregulation of p53 as in our studies further supports cell cycle arrest associated with senescence), and (b) evidence of concomitant upregulation of several components of the inflammatory secretory phenotype (SASP, Gorgoulis et al 2019 PMID: 31675495).

Our attempts to use β -gal staining as a measure of senescence in cultured cells *in vitro* in a wide variety of conditions resulted in uniformly stained cells where numbers of cells and levels of staining were indistinguishable in negative controls and positive controls. We observed similar outcomes in brain tissues. Considering that β -gal activity is not mechanistically involved in cellular senescence and does not provide definitive evidence of its activation, we measured the panel of cell cycle arrest and SASP factors that are robust and well-defined markers of senescence and mediators of the response (Gorgoulis et al 2019 PMID: 31675495), to document the activation of senescence with highest accuracy.

Before starting the studies reported, we consulted and discussed our approach for the detection of senescence Dr. Jim Kirkland, a recognized leader in senescence biology who contributed to establishing its markers.

R2 Comment #5. “ .. *Although eNOS activation and inhibition were studied by phosphorylation at S1177 and T495, NO production should be measured by DAF-FM staining and Griess assay.*”

We thank the reviewer for this comment and agree with her/his suggestion. We explored the important point raised by the reviewer by performing numerous experiments to measure nitric oxide release by both DAF-FM staining and the Griess assay using various formats. Unfortunately, DAF-FM was toxic to primary human brain endothelial cells (HBEC), leading to loss of cellular structure and detachment of cells at all concentrations tested. This precluded the use of DAF-FM for measurements of NO release.

We also ran numerous experiments where we stimulated either primary HBEC or bEnd.3 cells, an immortalized mouse brain endothelial cell line, with acetylcholine to promote NO release. Levels of nitrite measured with the Griess assay (Cat# STA-802 Cell Biolabs) were consistently below the detection limit in culture media from both HBEC and bEnd.3 cells, as well as in brain lysates from WT and P301S(PS19) mice in all conditions tested, even when stimulated with ACh (10 μ M cultured cells, 90 μ g/kg mice), or by shear stress. Levels of nitrate in cell media and in mouse brain tissues were also very low and highly variable even after ACh stimulation. Thus we were unable to directly measure NO release in relevant cell culture models or in brain tissues of WT or P301S(PS19) mice.

R2 Comment #6. “ .. *The authors indicated that microtubule destabilization induced by soluble tau aggregates might affect the transport of eNOS to membrane compartments. However, the evidence was not solid. It's better to provide visible evidence to show the translocation of eNOS.*”

We thank the reviewer for this important comment. As per the reviewer's suggestion, we ran additional experiments to measure eNOS localization in HBECs treated with vehicle, monomeric tau, or soluble aggregated tau using immunocytochemistry. Consistent with the observed decrease in eNOS activating phosphorylation and increased eNOS inhibitory phosphorylation (**Figure 2G-J** in the revised manuscript), we found that treatment of HBEC with tau monomers or with soluble aggregated tau significantly decreased the amount of eNOS present at the cell periphery in response to acetylcholine stimulation as compared to vehicle control. These data are now reported in **Figure 2 E,F** in the revised manuscript. The text of the Methods (page 19, line 482-492), Results (page 7, line 157-163) and Discussion (page 13, line 325-329) sections in the revised manuscript have been modified to describe these studies, as follows:

Methods (page 19, line 482-492)

“eNOS Localization: HBEC were plated in a 96-well plate and treated with vehicle (PBS, control), 40 μ g/mL monomeric tau (M. tau), or, 40 μ g/mL tau oligomers (O. Tau) for 42 hours. HBECs were then treated with media only or 10 μ M acetylcholine to induce eNOS translocation for 20 minutes. HBEC were fixed in PFA and immunostained with antibodies for eNOS and β -tubulin and stained with DAPI according to the methods described in the ICC methods, above. HBEC were imaged at 175x using Zoe Fluorescent Cell Imager (Bio-Rad Laboratories, Hercules, CA, USA). Images were analyzed using the plot profile function in Fiji ImageJ (NIH, Bethesda, MD, USA). eNOS localization within the cell was separated into nuclear, perinuclear, cytoplasmic, and edge regions as determined by the plot profile (gray value versus distance in pixels from the center of the nucleus to the edge of the cell) and visualization of features within each individual cell. Six samples from two separate biological replicates were analyzed for each group using Prism 8 (Graphpad, San Diego, CA, USA).”

Results (page 7, line 157-163)

“.. To determine if microtubule destabilization following soluble pathogenic tau uptake by HBEC inhibits eNOS translocation to the cell surface we measured eNOS localization in HBEC stimulated with ACh after exposure to soluble tau oligomers, tau monomers, or vehicle control. Consistent with the requirement for the microtubule cytoskeleton in the transport of eNOS 40,42, we found that eNOS abundance at the cell periphery was significantly decreased in HBEC treated with either monomeric or oligomeric tau, where the microtubule cytoskeleton is destabilized, as compared to controls ($p < 0.001$, Figure 2E-F).”

Discussion (page 13, line 325-329)

“...we found that eNOS translocation to the membrane of human brain endothelial cells was significantly reduced causing a significant decrease in phosphorylation-mediated eNOS activation and a significant increase in phosphorylation-mediated eNOS deactivation. Taken together, these data support the hypothesis that pathogenic tau destabilizes the microtubule cytoskeleton in human brain endothelial cells, inhibiting eNOS activation by disrupting transport of eNOS to membrane compartments.”

R2 Comment #7. “ .. *Tau oligomer-specific monoclonal antibody demonstrated protective effects against endothelial dysfunction and microvascular deficits. Does it alleviate pathological changes and symptoms of AD like cognitive impairment?*”

We thank the reviewer for this important question. Our experimental design precluded behavioral studies to measure memory, but prior studies from the laboratory of our coauthor Dr. Rakez Kaye, who developed TOMA and immunotherapy using this antibody, have demonstrated that TOMA treatment improves working memory (Y-maze task), sensory recognition memory (novel object recognition task), and contextual memory (fear conditioning task) in mouse models of Alzheimer's disease tauopathy including P301L(JNPL3) (Castillo-Carranza et al. 2014 PMC6608097), and hTau (Castillo-Carranza et al. 2014 PMID:24603946), and amyloidopathy (Castillo-Carranza et al. 2015 PMC6705372).

(3) Reviewer #3 (Remarks to the Author):

“Soluble pathogenic tau enters brain vascular endothelial cells and drives cellular senescence and brain microvascular dysfunction in tauopathy.

This paper builds on the author's previous demonstration that tau oligomers accumulate in microvascular endothelial cells in AD and other tauopathies. The current study shows similar findings in P201S tau mice and demonstrates that tau oligomers use a heparan sulfate proteoglycan dependent pathway to enter brain microvascular cells, which is followed by marked endothelial cell dysfunction including acquisition of the senescence associated secretory phenotype (SASP). The study provides novel information that is of broad interest.”

R3 Comment #1. “ .. *Please provide a higher power confocal view of Fig. 1b so that the location of the vascular accumulation of tau in P301S mice can be better ascertained. Electron micrographs would be particularly useful. Also, please provide a low power view of the entire brain section so that regional accumulation of vascular tau can be ascertained.*”

We thank the reviewer for this important suggestion. To address the reviewer's concern, we performed additional studies similar to those shown in the previous **Figure 1B**, now **Figure 1A**, where we collected both high magnification and low magnification confocal images of brain sections from WT and P301S(PS19) mice that were processed as in the studies show in **Figure 1A**. Data are shown in **Supplementary Figures 4C** and **4B** respectively in the revised manuscript. We modified the text of the Figure Legends section to include a description of these studies as follows:

Figure legends (page 42, line 983-987):

“B, Representative low-magnification images (10x) of cortical brain sections showing total tau (Tau5, green) immunoreactivity in P301S(PS19) and WT brain microvasculature (lectin, red) at 8 months of age. **C**, Representative high-magnification images (63x+2.5x digital zoom) of cortical brain sections showing total tau

total tau (Tau5, green) immunoreactivity P301S(PS19) and WT brain endothelial cells (CD31, red) at 8 months of age.”

R3 Comment #2. “ .. Does endothelial dysfunction require endogenous endothelial tau expression, as could be demonstrated through tau siRNA experiments? The data presented do not define a causal pathway, as both internalized tau as well as endogenous tau could contribute to reduced eNOS activation. “

We agree with the reviewer. While we show that internalized misfolded aggregated tau induces destabilization of the microtubule cytoskeleton and reduces eNOS activation (**Figure 2A-B** and **Figure 2 G-H**), it is possible that internalized misfolded aggregated tau could reduce eNOS activation in ways not dependent on aggregation of endogenous tau.

To define whether endogenous tau is required for destabilization of the microtubule cytoskeleton and reduction of eNOS activation triggered by entry of aggregated tau into primary human brain microvascular endothelial cells (HBEC) we set up siRNA experiments where we used 3 different validated siRNAs targeting human tau (Cat #4392420, ID #s8508, 8509, 8510, ThermoFisher) in primary human brain microvascular endothelial cells (HBEC). Similar to our observations in experiments using DAF-FM (see response to Comment #5, Reviewer #2) we found that the siRNA transfection reagent (lipofectamine) was toxic to primary HBEC at all concentrations that would result in effective transfection of siRNAs. Lipofectamine-induced toxicity thus precluded the use of siRNA-based approaches for these studies.

We next sought to use a lentivirus vector encoding an shRNA specific for human tau to knock down expression of tau in HBEC. Transduction of cells with lentiviral vectors requires the use of polybrene. Similar to our findings with DAF-FM and lipofectamine, we found that polybrene was toxic to HBEC at all concentrations tested. The use of lentivirus vectors to reduce endogenous human tau expression in HBEC was thus also precluded.

Last, we acquired *Mapt*^{-/-} mice to generate primary cultures of brain vascular endothelial cells null for the mouse tau gene. We have substantial experience isolating and culturing primary brain vascular endothelial cells from mice. The first three attempts at generating primary *Mapt*^{-/-} brain vascular endothelial cells from *Mapt*^{-/-} mice, however, were unsuccessful, with almost no cells surviving after plating. These studies, that are performed in 5 month-old mice (the age at which brain microvasculature is fully mature and can be used to generate primary brain vascular endothelial cells) are still ongoing and will continue in the future.

R3 Comment #3. “.. SASP markers are different *in vitro* vs *in vivo*, with MCP-1 being the most upregulated *in vivo* with no change in IL-6, whereas the opposite pattern is observed *In vitro* where IL-6 is upregulated ~25 fold. The authors suggest that a lower incidence of *in vivo* uptake and the influence of other cell types in the *in vivo* neurovascular unit may explain the discrepancy. This is unsatisfactory as a lower incidence should reduce the levels of SASP markers but not alter the pattern of activation, and the influence of other cells types has not been experimentally addressed, such as with transwell assays that include pericytes and astrocytes.”

We thank the reviewer for this comment. We agree with the reviewer that the rationale provided in the first version of our manuscript to explain the discrepancy between the SASP profile in cultured cells and in purified microvasculature from P301S(PS19) mice was not entirely satisfactory.

The SASP composition and strength varies substantially among cell types (Gorgoulis et al 2019 PMID:31675495) and also within a given cell type depending on multiple factors, including prominently upstream signals from its milieu (Acosta et al 2013 PMC3732483, Gorgoulis et al 2019 PMID:31675495, Davalos et al 2013 PMC3653366, Li and Chen, 2018, Childs et al 2015 PMC4748967).

In our studies, primary human brain endothelial cells (HBEC) were exposed to a single, homogeneous dose of purified soluble tau aggregates in defined culture conditions *in vitro*. In contrast, mouse brain endothelial cells *in vivo* were embedded in the diverse multicellular niche of the brain microvasculature where, in addition to pathogenic soluble tau aggregates, they were continuously exposed to other cell types and to a host of substances present in the brain interstitial fluid, including those specific to the disease state in P301S(PS19) mice undergoing tauopathy. Thus, the differences that we observed in the profile of inflammatory factors secreted by HBEC undergoing senescence *in vitro* as compared to that of mouse endothelial cells undergoing

senescence *in vivo* may arise from substantial differences in the environment in which these cells were embedded while undergoing pathogenic tau-induced senescence. Potential differences in SASP composition across different species (human endothelial cells *in vitro*, mouse endothelial cells *in vivo*) may also contribute to differences in SASP profile. The text of the Results (page 10, line 243-247) and the Discussion (page 15, line 368-382) sections of the revised manuscript have been modified to include the information above, as follows:

Results (page 10, line 243-247)

“The composition of SASP in microvasculature purified from brains of P301S(PS19) mice was different from that in primary brain vascular endothelial cells exposed to pathogenic soluble tau *in vitro*, likely due to profound differences in environment make-up *in vivo* in brain microvasculature vs. *in vitro* in monotypic cell culture conditions (Gorgoulis et al 2019 PMID:31675495) and potentially also species-specific differences (human endothelial cells *in vitro*, mouse endothelial cells *in vivo*).”

Discussion (page 15, line 371-382)

“It has been shown that the SASP composition and strength varies substantially among cell types (Gorgoulis et al 2019 PMID:31675495) and also within a given cell type depending on multiple factors, including prominently upstream signals from its milieu (Acosta et al 2013 PMC3732483, Gorgoulis et al 2019 PMID:31675495, Davalos et al 2013 PMC3653366, Li and Chen, 2018, Childs et al 2015 PMC4748967). Cultured human brain endothelial cells were exposed to a single, relatively homogeneous dose of purified soluble tau aggregates in defined cell culture conditions *in vitro*. In contrast, mouse brain endothelial cells undergoing senescence *in vivo* were embedded in the diverse multicellular niche of the brain microvasculature where, in addition to pathogenic soluble tau aggregates, they were continuously exposed to other cell types and to a host of substances present in the brain interstitial fluid, including those specific to the disease state in P301S(PS19) mice undergoing tauopathy. Thus, the differences in profile of inflammatory factors secreted during senescence by human brain endothelial cells *in vitro* as compared to that of mouse brain endothelial cells undergoing senescence *in vivo* may arise from numerous differences in the environmental milieu that these cells were exposed to while undergoing pathogenic tau-induced senescence.”

We also thank the reviewer for suggesting co-culture studies where the impact of different cells of the neurovascular unit on endothelial cell senescence, and the converse, can be ascertained. We have already established a transwell system for astrocytes and neurons and are developing a similar model using endothelial cells undergoing senescence induced by pathogenic tau co-cultured in a transwell system with astrocytes or smooth muscle cells. We expect that these studies will be reported in a future, separate publication.

R3 Comment #4. *“.. The immunotherapy experiments do not provide mechanistic insights. For example, it is possible that endothelial function is restored indirectly via improved neuronal function, neurovascular coupling, and vascular reactivity. Also, TOMA treatment would also reduce tau found in the blood, and the peripheral vs CNS contributions of tau on endothelial cell dysfunction cannot be distinguished by the present experiments. These limitations should be acknowledged.”*

We agree with the reviewer that immunotherapy could have effects on multiple cell types in addition to endothelial cells, and that peripheral vs. central contributions of tau on endothelial cell dysfunction are not distinguished by our studies. We have added a statement to the Results section in the revised manuscript where this limitation is acknowledged (page 12, line 284-290), as follows:

Results section (page 12, line 284-290)

“ .. These data indicate that soluble aggregated tau has causal role in the etiology of microvascular dysfunction in P301S(PS19) mice. A limitation, however, is that TOMA is expected to reduce levels of soluble tau aggregates in multiple brain cell types, and potentially in the periphery. Thus, whether the restoration of microvascular function by TOMA is driven by the removal of soluble tau aggregates from microvascular endothelial cells alone, or in combination with removal of pathogenic tau from other brain cell types or potentially from cell types of the periphery cannot be ascertained from our data.”

Clearance of tau from brain by TOMA, however, is associated with and evidenced by a marked *increase* (not a reduction) in plasma levels of tau (Castillo-Carranza et al 2014 PMC6608097, Yanamandra et al 2017 PMC5727571). Like for other antibody-antigen complexes in circulation, tau bound to TOMA is expected to be cleared by the liver, limiting the potential for antibody-bound tau to have a biological impact on peripheral organs.

R3 Comment #5. "Methods are appropriate and sufficiently detailed. With respect to statistical methods, please provide additional information for meeting ANOVA criteria (i.e. normality) when this test is used."

We thank for the reviewer for noting this omission. The text of the Methods section (page 23, line 597-600) in the revised manuscript has been modified to include a description of the statistical analyses used to determine if variances were equal among experimental groups, as follows:

Methods section (page 23, line 597-600)

" .. Normality was tested using the Brown-Forsythe analysis, when variances were unequal, a Kruskal-Wallis test was performed followed by a Dunn's multiple comparison's test. $p < 0.05$ was considered significant. The Grubbs' test was used to identify statistical outliers ($\alpha = 0.05$)."

REVIEWER COMMENTS

Reviewer #1 (Remarks to the Author):

The manuscript entitled, " Soluble pathogenic tau enters brain vascular endothelial cells and drives cellular senescence and brain microvascular dysfunction in tauopathy" explores the role of soluble pathogenic tau in mediating microvascular dysfunction. The role of endothelial dysfunction of eNOS dysfunction also investigated. I have reviewed the revised manuscript meticulously and authors have addressed all the issued raised by reviewers. Thus, I would like to recommend this manuscript for publication.

Reviewer #4 (Remarks to the Author):

The work by Hussong et al. investigates the role of soluble pathogenic forms of tau in mediating endothelial dysfunction and senescence. The authors showed that soluble tau aggregates were are internalized by brain microvascular endothelial cells, promoting phosphorylation of endogenous tau. Tau internalization in endothelial cells alters both subcellular localization and phosphorylation of eNOS eNOS and induced expression of genes associated with endothelial senescence. In addition, removal of soluble tau aggregates by tau oligomer-specific monoclonal antibody restored endothelial function. The authors addressed most of the concerns and suggestions of previous reviewers. However, I think there are still questions that need to be addressed.

One major limitation of the study is the lack of evidence that ability of endothelial cells to release NO is impaired after administration of exogenous tau. The authors unsuccessfully tried to measure NO release by using both DAF-FM and Griess assay in both primary human brain endothelial cells (HBEC) and bEnd.3 cell line. The lack of success with bEnd.3 cells is surprising since it has been shown that Ach or A23187 increases NO production to levels that can be detected by both DAF-FM and Griess assay (Govers et al., *Biochem. J.* 361: 193, 2002). The authors should consider using A23187 or increasing the concentration and/or the incubation time of Ach. In addition, the authors should consider that level of confluence of bEnd.3 cells might also affect the ability to produce NO. The addition of data on the effect of tau on NO release would strengthen the major findings of the study.

On the same line, subcellular localization of eNOS should be better characterized by performing subcellular protein fractionation. Alternatively, since it has been shown that eNOS dissociates from Cav-1 when active (Garcia-Cardena G et al., *J Biol Chem.* 272:25437, 1997; Ju H et al., *J Biol Chem.* 272:18522, 1997; Feron et al., *J Biol Chem.* 273:3125, 1998), the authors could test the effect of tau on the association between eNOS and Cav-1 by co-IP. Furthermore, the authors showed and stated in the discussion that "eNOS translocation to the membrane was significantly reduced causing a significant decrease in phosphorylation-mediated eNOS activation and a significant increase in phosphorylation-mediated eNOS deactivation". However, to my knowledge, eNOS phosphorylation is increased whereas eNOS activity is reduced when the enzyme is bound to membranes (Ramadoss et al., *Hypertension* 59:1052, 2012). It has been shown that eNOS activation is associated with translocation from the membrane to the cytosol and other subcellular organelles and not viceversa (Prabhakar P et al., *J Biol Chem.* 273:27383, 1998). Therefore, if tau reduces eNOS activity, I would expect eNOS to be found more associated to the plasma membrane.

Another concern regards the cerebrovascular studies. The vasodilatory response to Ach is almost absent in WT mice, suggesting that this might not be the most suited model to test the effect of tau on brain endothelial function in vivo. Such small responses make the interpretation of the data extremely difficult, including the potential beneficial effect of TOMA administration. In light of this, as mentioned above, being able to show that tau reduces NO release in brain endothelial cell cultures would greatly improve the study. Alternatively, the authors could try to test the effect of exogenous tau administration on the endothelial response to Ach in C57Bl6 mice where the vasodilatory response seems to be more consistent. The authors could test whether Tau is picked up

by endothelial cells and whether the endothelial function is altered.

Minor points: in Figure 1A, Tau5 is present in endothelial cells but not in neurons of WT mice. This is surprising. In addition, neuronal staining seems to be absent in PS19 as well. A lower magnification image of the brain parenchyma might help. In addition, orthogonal views of the staining could better show the colocalization between Tau22 and/or Tau5 and the vascular markers.

How do the authors reconcile these findings with the increasing evidence suggesting that vascular alterations such as blood-brain barrier disruption or endothelial dysfunction might take place long before the deposition of A β or the accumulation of phosphorylated tau? These data would suggest that tau pathology might promote vascular alterations and not vice versa.

Response to the Reviewers

Reviewer #1 (Remarks to the Author):

Comment: "The manuscript entitled, " Soluble pathogenic tau enters brain vascular endothelial cells and drives cellular senescence and brain microvascular dysfunction in tauopathy" explores the role of soluble pathogenic tau in mediating microvascular dysfunction. The role of endothelial dysfunction of eNOS dysfunction also investigated. I have reviewed the revised manuscript meticulously and authors have addressed all the issues raised by reviewers. Thus, I would like to recommend this manuscript for publication."

Response: We thank the reviewer for her/his recommendation for publication.

Reviewer #4 (Remarks to the Author):

Comment #1: "The work by Hussong et al. investigates the role of soluble pathogenic forms of tau in mediating endothelial dysfunction and senescence. The authors showed that soluble tau aggregates were internalized by brain microvascular endothelial cells, promoting phosphorylation of endogenous tau. Tau internalization in endothelial cells alters both subcellular localization and phosphorylation of eNOS and induced expression of genes associated with endothelial senescence. In addition, removal of soluble tau aggregates by tau oligomer-specific monoclonal antibody restored endothelial function. The authors addressed most of the concerns and suggestions of previous reviewers."

Response #1: We thank the reviewer for her/his comment.

Comment #2: "However, I think there are still questions that need to be addressed. One major limitation of the study is the lack of evidence that ability of endothelial cells to release NO is impaired after administration of exogenous tau. The authors unsuccessfully tried to measure NO release by using both DAF-FM and Griess assay in both primary human brain endothelial cells (HBEC) and bEnd.3 cell line. The lack of success with bEnd.3 cells is surprising since it has been shown that Ach or A23187 increases NO production to levels that can be detected by both DAF-FM and Griess assay (Govers et al., Biochem. J. 361: 193, 2002). The authors should consider using A23187 or increasing the concentration and/or the incubation time of Ach. In addition, the authors should consider that level of confluence of bEnd.3 cells might also affect the ability to produce NO. The addition of data on the effect of tau on NO release would strengthen the major findings of the study."

Response #2: We agree with the reviewer that showing that tau treatment effects nitric oxide production in cultured endothelial cells strengthens the major findings of our study. To address this important point, we further optimized our protocols for measuring nitric oxide using DAF-FM. As the reviewer had suggested, changing our culture conditions we allowed us to measure NO production in primary human brain endothelial cells (HBEC) using this reagent. Our data show that both oligomeric tau and monomeric tau significantly inhibit NO production in HBEC as measured by DAF-FM ($p < 0.0001$ and $p < 0.01$, respectively). These data are discussed in the Results section (page 8), in the Discussion section (page 13-14), methods (page 21), and are shown in **Figure 2H** of the revised manuscript. Changes in the revised manuscript have been highlighted in yellow.

Comment #3: "On the same line, subcellular localization of eNOS should be better characterized by performing subcellular protein fractionation. Alternatively, since it has been shown that eNOS dissociates from Cav-1 when active (Garcia-Cardena G et al., J Biol Chem. 272:25437, 1997; Ju H et al., J Biol Chem. 272:18522, 1997; Feron et al., J Biol Chem. 273:3125, 1998), the authors could test the effect of tau on the association between eNOS and Cav-1 by co-IP."

Furthermore, the authors showed and stated in the discussion that “eNOS translocation to the membrane was significantly reduced causing a significant decrease in phosphorylation-mediated eNOS activation and a significant increase in phosphorylation-mediated eNOS deactivation”. However, to my knowledge, eNOS phosphorylation is increased whereas eNOS activity is reduced when the enzyme is bound to membranes (Ramadoss et al., Hypertension 59:1052, 2012). It has been shown that eNOS activation is associated with translocation from the membrane to the cytosol and other subcellular organelles and not viceversa (Prabhakar P et al., J Biol Chem. 273:27383, 1998). Therefore, if tau reduces eNOS activity, I would expect eNOS to be found more associated to the plasma membrane.”

Response #3: The reviewer is correct that literature has conflicting reports as to where active eNOS is located. Hecker et al. (Biochem J. 299:247-252, 1994), reported that there are two major pools of activated eNOS: One associated with the cell membrane and one associated with the endoplasmic reticulum. Govers et al. (Biochem J. 361:193-201, 2002) also reported that eNOS activation is associated with its presence at cell-cell contacts; this would explain why cultures of endothelial cells need to be close to confluency to observe substantial eNOS activation. It has also been reported that while eNOS becomes unbound from caveolin for activation, the enzyme remains at the cell membrane, is phosphorylated at S1177, and interacts with calmodulin and Hsp90 to become active (Fleming, Eur J Physiol 459:793-806, 2010; Carver and Schnitzer, Nature Reviews Cancer 3: 571-581, 2003). Our data shows that the active pool of eNOS at the cell membrane is significantly reduced in HBEC treated with pathogenic oligomeric tau and stimulated with acetylcholine (cell edge-associated signal, **Figure 2E-F**). We also found an apparent reduction in eNOS localized to the endoplasmic reticulum (perinuclear compartment, **Figure 2G** in the revised manuscript) although this apparent difference is not significant. The impact of pathogenic oligomeric tau on eNOS localization in HBEC is consistent with the observed decrease in NO production (**Figure 2H** in the revised manuscript), reduction in active, S1177-phosphorylated eNOS (**Figure 2I-J**), and increase in eNOS phosphorylated at its inhibitory site, T485 (**Figure 2K-L**) arising from pathogenic tau exposure. As the reviewer rightly notes, other studies (Ramadoss et al., Hypertension 59:1052, 2012, Prabhakar P et al., J Biol Chem. 273:27383, 1998) have suggested that activation of eNOS is associated with its translocation from the membrane to the cytosol. It is possible that these apparent conflicts in the literature may reflect cell subtype-specific and potentially also cell state-specific differences in localization of eNOS-interacting proteins required for its activation (e.g. localization of pools of various interaction partners of the enzyme) or in other factors present in different types of endothelial cells (e.g. cardiac endothelial cells vs. brain endothelial cells, activated vs. quiescent endothelial cells).

To address this point, we have added the text below to the Discussion section of the revised manuscript (Page 14 line 331-348), as follows:

“ .. Conflicting data have been reported for studies that examined the subcellular location of active forms of eNOS. Hecker et al. reported that there are two major pools of activated eNOS: one associated with the cell membrane and one associated with the endoplasmic reticulum (Biochem J. 299:247-252, 1994). Govers et al. (Biochem J. 361:193-201, 2002) also reported that eNOS activation is associated with its presence at cell-cell contacts; this would explain why cultures of endothelial cells need to be close to confluency to observe substantial eNOS activation. It has also been reported that while eNOS becomes unbound from caveolin for activation, the enzyme remains at the cell membrane, is phosphorylated at S1177, and interacts with calmodulin and Hsp90 to become active (Fleming, Eur J Physiol 459:793-806, 2010; Carver and Schnitzer, Nature Reviews Cancer 3: 571-581, 2003). Our data shows that treatment with pathogenic, oligomeric tau decreases levels of eNOS at the cell membrane in HBEC stimulated with acetylcholine. We also found an apparent reduction in eNOS localized to the endoplasmic reticulum, although this difference was not significant. The impact of pathogenic oligomeric tau on eNOS localization in HBEC is consistent with the observed decrease in NO production, reduction in active, S1177-phosphorylated eNOS, and increase in eNOS phosphorylated at its inhibitory site, T485, arising from pathogenic tau exposure. Other reports however, have suggested that activation of eNOS is associated with its translocation from the membrane to the cytosol (Ramadoss et al., Hypertension 59:1052, 2012, Prabhakar P et al., J Biol Chem. 273:27383, 1998). It is possible that these apparent conflict in the literature may reflect endothelial cell subtype-specific and potentially also cell state-specific differences in localization of eNOS-interacting proteins required for its activation (e.g. localization of pools of various interaction partners of the enzyme) or

in other factors present in different types of endothelial cells (e.g. cardiac endothelial cells vs. brain endothelial cells, activated vs. quiescent endothelial cells).

Comment #4: *“Another concern regards the cerebrovascular studies. The vasodilatory response to Ach is almost absent in WT mice, suggesting that this might not be the most suited model to test the effect of tau on brain endothelial function in vivo. Such small responses make the interpretation of the data extremely difficult, including the potential beneficial effect of TOMA administration. In light of this, as mentioned above, being able to show that tau reduces NO release in brain endothelial cell cultures would greatly improve the study. Alternatively, the authors could try to test the effect of exogenous tau administration on the endothelial response to Ach in C57Bl6 mice where the vasodilatory response seems to be more consistent. The authors could test whether Tau is picked up by endothelial cells and whether the endothelial function is altered.”*

Response #4: We thank the reviewer for bringing up this important point. We have tried administration of pathogenic soluble tau in brains of WT C57BL/6J brains extensively. Administration of pathogenic soluble tau in all instances resulted in rapid detection and engulfment of tau by local activated microglia within hours of treatment, precluding uptake of exogenous pathogenic tau by any other brain cell types, including endothelial cells.

As the reviewer points out, wild-type animals in the hybrid B6:C3H background of the P301S(PS19) mouse model of tauopathy show reduced responses to acetylcholine with regards to C57BL/6J mice. Genetic background-related differences in vascular responses have been reported in the past (Kang et al., JCBFM 35:912-916, 2015, Barone et al., JCBFM 13:683-692, 1993). While the magnitude of baseline endothelium-dependent vascular response to acetylcholine of wild-type animals in the B6:C3H background is reduced as compared to those of C57BL/6J mice (which demonstrate the effectiveness of our experimental approach), we documented profound and significant impairment of endothelium-dependent vascular responses in transgenic P301S(PS19) mice expressing high levels of pathogenic oligomeric tau as compared to B6:C3H controls (**Figures 4A and 6A**). These changes are accompanied by decreased brain vascular microtubule stability (**Figure 5A-B**), decreased eNOS activation (**Figure 5C-D**), increased eNOS inhibition (**Figure 5E-F**), and increased markers of cellular senescence (**Figure 5 G-H**) in brain microvasculature of P301S(PS19) mice, which are one of the best-characterized models of tauopathy available. These outcomes are recapitulated in cultured HBEC exposed to purified pathogenic oligomeric forms of tau, and are partially negated by the selective removal of pathogenic oligomeric tau from P301S(PS19) mice using immunotherapy. Because high levels of pathogenic soluble tau cause profound deficits in vascular reactivity in P301S(PS19) mice, our data strongly support a role of pathogenic soluble tau in the etiology of endothelium-dependent vascular dysfunction in this widely used and well-characterized model of tauopathy.

Comment #5: *“Minor points: in Figure 1A, Tau5 is present in endothelial cells but not in neurons of WT mice. This is surprising. In addition, neuronal staining seems to be absent in PS19 as well. A lower magnification image of the brain parenchyma might help. In addition, orthogonal views of the staining could better show the colocalization between Tau22 and/or Tau5 and the vascular markers.”*

We thank the reviewer for this comment and agree that neuronal tau staining in brains of WT mice in **Figure 1A** is practically invisible in the low-quality versions of immunohistochemistry images, that we were required to include in our original submission. In those low-quality images, neuronal tau signal is not prominent for WT animals. Neuronal tau in WT mouse brains, however, is evident in high quality images that we generated and will submit for publication. These images are presented in the revised manuscript, and are included below.

Comment #6: “How do the authors reconcile these findings with the increasing evidence suggesting that vascular alterations such as blood-brain barrier disruption or endothelial dysfunction might take place long before the deposition of A β or the accumulation of phosphorylated tau? These data would suggest that tau pathology might promote vascular alterations and not vice versa.”

We thank the reviewer for this important comment. The reviewer is correct that large studies in humans have suggested that vascular defects precede both amyloid-beta and phosphorylated tau accumulation. Our studies do not address this observation but suggest that, similar to amyloid-beta, the accumulation of pathogenic oligomeric forms of tau exacerbate vascular dysfunction, contributing together with amyloid-beta to establish a feed-forward loop of vascular damage and pathogenic amyloid-beta and oligomeric tau accumulation. While we did not observe changes in blood-brain barrier permeability as measured by fibrinogen extravasation in P301S(PS19) mice at the ages tested (**Figure S4E**), our studies show significant damage to endothelial cell function (**Figures 4, 5, 6**) as well as smooth muscle cell function (**Figure S3**) in P301S(PS19) mice. To address this important point, we have added the text below to the Discussion section of the revised manuscript (Page 16, line 381-389), as follows:

“Large studies in human cohorts (Iturria-Medina et al., 2016, Nat Commun) have suggested that vascular defects precede both amyloid-beta and phosphorylated tau accumulation. While our studies do not address this aspect of AD pathogenesis, our data suggest that, similar to amyloid-beta, the accumulation of pathogenic oligomeric forms of tau exacerbate vascular dysfunction, contributing together with amyloid-beta to establish a feed-forward loop of vascular

damage and pathogenic amyloid-beta and oligomeric tau accumulation during progression of AD. While we did not observe changes in blood-brain barrier permeability as measured by fibrinogen extravasation in P301S(PS19) mice at the ages tested, our studies show significant damage to endothelial cell function as well as smooth muscle cell function caused by pathogenic oligomeric forms of tau in P301S(PS19) mice.”

REVIEWER COMMENTS

Reviewer #4 (Remarks to the Author):

The authors addressed most of my concerns. However, although now better supported by in vitro data, I still believe that the data on cerebrovascular reactivity are not solid. The in vivo evidence that tau might selectively impair endothelial function is still rather weak. The authors have shown that eNOS phosphorylation status is altered in PS19 mice. However, there is still no sound evidence that eNOS activity is reduced in vivo. As I previously mentioned the cerebrovascular studies are not entirely convincing. Acetylcholine (ACh) induces a 1-2% increase in cerebral blood flow in control mice. Considering the noise associated with the laser-doppler signal, such increase is small. On the same line, the reduction in the CBF increase induced by ACh in PS19 mice (<5% at 4months of age) is also extremely small. Also, how would be the mechanism by which ACh mediates a reduction in CBF in PS19?

In addition, the experiment with sodium nitroprusside (SNP) suggests that vascular smooth muscle cells (VSMCs) function may be altered in PS19 mice (10 months of age). More precisely, the data suggest that sensitivity of VSMCs to NO is altered in PS19 mice. Therefore, the small effect on the ACh-dependent vasodilation found in PS19 mice could potentially be dependent on a reduced sensitivity of VSMCs to NO and not to reduced endothelial NO production in vivo. Therefore, to exclude that reduced VSMCs-NO sensitivity does not play a role, the authors should test the effect of other VSMCs relaxants such as adenosine. In addition, the concentration of SNP used by the authors to induce endothelium-independent vasodilation is much higher than the concentration normally used in cranial window studies (i.e. Sobey and Faraci, *Stroke*, 28:837, 1997, Didion et al, *Arterioscler Thromb Vasc Biol*, 27:2576, 2007). Considering that the SNP-induced vasodilation is reduced with such high concentration, the data suggest that VSMCs function might be severely altered in PS19 mice. Based on these considerations, the conclusion made by the authors that "the vascular changes are due to an impairment in endothelial function which is of a greater magnitude than the dysfunction induced on VSMCs" is not supported by the data. By using lower concentrations (0.1-100µM), the authors could find an even greater reduction in VSMCs-dependent vasodilation further suggesting that VSMCs function is severely altered in PS19 mice. The data are in contrast with recent data showing that endothelial and VSMCs function are not altered in young PS19 mice (*Nat Neurosci*, 23: 1079, 2020). Therefore, it would also be interesting to know whether VSMCs function is also altered in younger mice. Similarly, it would be interesting to know whether tau selectively alters eNOS activity or more profoundly alters endothelial function. To this end, the authors could test agents (bradykinin, A23187, etc) that induce endothelium-dependent vasodilation with mechanisms other than eNOS activation.

Response to the reviewer

All changes to the manuscript have been highlighted in yellow.

Comment #1: *The authors addressed most of my concerns. However, although now better supported by in vitro data, I still believe that the data on cerebrovascular reactivity are not solid. The in vivo evidence that tau might selectively impair endothelial function is still rather weak. The authors have shown that eNOS phosphorylation status is altered in PS19 mice. However, there is still no sound evidence that eNOS activity is reduced in vivo. As I previously mentioned the cerebrovascular studies are not entirely convincing. Acetylcholine (ACh) induces a 1-2% increase in cerebral blood flow in control mice. Considering the noise associated with the laser-doppler signal, such increase is small. On the same line, the reduction in the CBF increase induced by ACh in PS19 mice (<5% at 4months of age) is also extremely small.*

We thank the reviewer for her/his comment, and would like to respectfully disagree. Genetic background-related differences in vascular responses have been reported in the past (Kang et al., JCBFM 35:912-916 PMC4640258; Barone et al., JCBFM 13:683-692, PMID 8314921). Transgenic (Tg) P301S(PS19) and WT littermate mice are in a unique hybrid B6:C3H background, generated by breeding P301S(PS19) Tg mice to WT littermates for many generations since the establishment of the line.

Our data indicate that diminished responses to ACh (2.5% and 3% increase in CBF, **Figure 4A and Figure 6A** in the revised manuscript) are a feature of the unique hybrid B6:CH3 background of P301S(PS19) mice, since simultaneous experiments using identical experimental tools showed a 9±2% CBF increase in C57BL/6J WT mice, a genetic background that is frequently used in studies of cerebrovascular function.

That the ACh-elicited responses in B6:CH3 WT mice are of smaller magnitude than those of C57BL/6J mice does not necessarily diminish their relevance. Albeit lower in magnitude than WT C57BL/6J, CBF responses to ACh in B6:CH3 WT mice are of low interindividual variability, are consistent across ages, and are significantly higher than those of congenic P301S(PS19) Tg littermate mice (Figure 4A in the revised manuscript). The studies of Figure 4A thus demonstrate both effectiveness of ACh in inducing vasodilation, and effectiveness of our experimental approach.

Irrespective of the magnitude of the vascular response to ACh in P301S(PS19) WT mice, our data demonstrate a profound and progressive impairment in endothelium-dependent vascular responses to ACh in P301S(PS19) Tg mice as compared to their WT controls (**Figure 4A and 6A** in the revised manuscript). As described in page 9 line 205, paradoxical vasoconstriction in response to ACh stimulation can arise from direct ACh action on VSMC in conditions of severe endothelial dysfunction (PMID 27936390, 15086237, 27348596, 23102612, 9736128, 3093861 and 1953614). Our studies show paradoxical vasoconstriction in response to ACh stimulation, reaching a magnitude of 8±2% reduction in CBF, indicative of severely diminished endothelial cell function in 8 month-old P301S(PS19) Tg mice (**Figure 4A** in the revised manuscript).

Further, to provide an independent direct measure of endothelial cell function in P301S(PS19) mice we performed new experiments that measured NO production in microvessels freshly isolated from P301S(PS19) Tg mice and NTg WT littermate controls. These studies (included as **Supplemental Figure 3** in the revised manuscript) show that microvessels from 9 month-old P301S(PS19) Tg mice produce significantly less nitric oxide than those of age-matched WT control mice. These data provide direct, independent evidence of endothelial cell dysfunction in P301S(PS19) Tg mice modeling tauopathy and are in agreement with our *in vivo* ACh studies (**Figure 4A and Supplemental Figure 3** in the revised manuscript). The data from our *ex vivo* studies of NO production in purified microvessels have been added to the revised manuscript as **Supplemental Figure 3** and are discussed on page 9, line 216 of the Results section. The Methods section has been edited accordingly (page 21, line 527).

Comment #2: Part 1: Also, how would be the mechanism by which ACh mediates a reduction in CBF in PS19?

We are grateful to the reviewer for her/his comment. We describe the mechanism by which ACh stimulation mediates a reduction in CBF in page 9, line 205 of the revised manuscript, as follows: *“Reduced or absent endothelial release of NO in conditions of severe endothelial dysfunction can lead to paradoxical vasoconstriction and vasospasm in response to ACh stimulation (PMID 27936390, 15086237, 27348596, 23102612, 9736128, 3093861, 1953614). ... Tauopathy arising from overexpression of human tau ... profoundly impaired endothelial responses in P301S(PS19) mice, such that ACh stimulation resulted in paradoxical vasoconstriction, likely as a result of direct action of ACh on vascular smooth muscle cells in the near absence of endothelium-dependent contributions.*

Comment #2: Part 2: *In addition, the experiment with sodium nitroprusside (SNP) suggests that vascular smooth muscle cells (VSMCs) function may be altered in PS19 mice (10 months of age). More precisely, the data suggest that sensitivity of VSMCs to NO is altered in PS19 mice. Therefore, the small effect on the ACh-dependent vasodilation found in PS19 mice could potentially be dependent on a reduced sensitivity of VSMCs to NO and not to reduced endothelial NO production in vivo. ... In addition, the concentration of SNP used by the authors to induce endothelium-independent vasodilation is much higher than the concentration normally used in cranial window studies (i.e. Sobey and Faraci, Stroke, 28:837, 1997, Didion et al, Arterioscler Thromb Vasc Biol, 27:2576, 2007). Considering that the SNP-induced vasodilation is reduced with such high concentration, the data suggest that VSMCs function might be severely altered in PS19 mice. Based on these considerations, the conclusion made by the authors that “the vascular changes are due to an impairment in endothelial function which is of a greater magnitude than the dysfunction induced on VSMCs” is not supported by the data. By using lower concentrations (0.1-100µM), the authors could find an even greater reduction in VSMCs-dependent vasodilation further suggesting that VSMCs function is severely altered in PS19 mice. The data are in contrast with recent data showing that endothelial and VSMCs function are not altered in young PS19 mice (Nat Neurosci, 23: 1079, 2020). Therefore, it would also be interesting to know whether VSMCs function is also altered in younger mice. Similarly, it would be interesting to know whether tau selectively alters eNOS activity or more profoundly alters endothelial function. To this end, the authors could test agents (bradykinin, A23187, etc) that induce endothelium-dependent vasodilation with mechanisms other than eNOS activation.*

We thank the reviewer for his/her comments. Based on their comments, we reviewed the experimental design for the SNP studies that were reported in the prior version of the manuscript. We found that, in contrast to all other animals used in the studies reported, P301S(PS19) mice that were used to test VSMC function with SNP were directly obtained from Jackson Laboratories (JAX) as retired breeders at 7-8 months of age. This was necessary because at that moment breeding animals from our own colony to perform SNP studies would have delayed our studies by at least 15 months.

Discrepant environmental conditions in different facilities and housing conditions can substantially impact phenotypes, including prominently AD-like phenotypes (Ryman and Lamb, PMID 17168645; Kaur et al PMC8469717; Colombo PMC8043748 microbiota). The animals used in the SNP flowmetry studies reported in the prior revised version of our manuscript were generated and housed in a JAX breeding colony, and used in flowmetry studies in transient housing outside of the barrier facility shortly after arrival, thus in drastically different conditions than animals generated our colony, including but not limited to breeder status. Because of these differences, we hypothesized that the outcomes measured in retired JAX breeders may not have adequately informed the biology of the animals that we used to generate all other data that we report.

To test this hypothesis we performed new experiments using P301S(PS19) mice generated and tested in our mouse colony, thus most comparable to animals that were used other studies reported in our manuscript. In contrast to previous SNP experiments on retired P301S(PS19) breeders from JAX, our studies using 9 month-old P301S(PS19) animals generated our colony demonstrated a significant increase in CBF stimulated by 20 μ M SNP (an SNP dose in the range of concentrations used in similar studies, 0.1-100 μ M) that was indistinguishable from that of WT mice (**Supplementary Figure 4** in the revised manuscript).

Thus, and in agreement with prior studies by other laboratories (Park et al. *Nat Neurosci* 2020 23:1079), our data rule out an involvement of VSMC dysfunction in the impaired response of P301S(PS19) mouse microvasculature to ACh, and indicate that VSMC are not negatively impacted by tauopathy even at late stages of disease in P301S(PS19) mice.

The text of the Results (page 10, line 227), Methods (page 25, line 630) and Discussion (page 15, line 362) have been modified to reflect these data.

We also agreed with the reviewer that more extensive testing of endothelial dysfunction in P301S(PS19) mice would strengthen the studies reported. However, because bradykinin and the Ca²⁺ ionophore A23187 induce vasodilation through mechanisms that include NO release (Ihara et al PMC1571924; Honing PMID:10856283; Chen et al PMC5935069), we would not be able to unequivocally distinguish an NO-dependent versus a non-NO-dependent deficit in bradykinin- and/or A23187-evoked vasodilation in P301S(PS19) mice.

Thus, to specifically address the reviewer's concern that the defect in endothelium-dependent NO release documented by flowmetry studies using ACh are driven by decreased NO, we performed additional studies to provide an independent direct measure of changes in NO-driven vasodilation in P301S(PS19) mice. These new experiments measured NO production in microvessels freshly isolated from P301S(PS19) Tg mice and NTg WT littermate controls using DAF-FM (please see response to Comment #1).

These new studies, presented in **Supplemental Figure 3** of the revised manuscript, show that microvessels from 9 month-old P301S(PS19) Tg mice produce significantly less NO than those of age-matched WT control mice. These data provide direct independent evidence of NO deficit-driven brain microvascular dysfunction in P301S(PS19) Tg mice modeling tauopathy, in agreement with *in vivo* ACh studies (**Figure 4A** in the revised manuscript).

Taken together with our new studies measuring VSMC function as responsiveness to an NO donor in P301S(PS19) versus WT littermate mice (**Supplementary Figure 4** in the revised manuscript), these data provide direct evidence of endothelial cell, but not VSMC dysfunction, in P301S(PS19) Tg mice modeling tauopathy.

REVIEWERS' COMMENTS

Reviewer #4 (Remarks to the Author):

The authors addressed my concerns.